# MarS: a Financial Market Simulation Engine Powered by Generative Foundation Model

**Junjie Li,**[*] **Yang Liu,**[*] **Weiqing Liu,**[†] **Shikai Fang, Lewen Wang, Chang Xu & Jiang Bian**
Microsoft Research Asia
`{junli,yangliu2,weiqing.liu,fangshikai,lewen.wang,`
`chanx,jiang.bian}@microsoft.com`

## Abstract

Generative models aim to simulate realistic effects of various actions across different contexts, from text generation to visual effects. Despite significant efforts to build real-world simulators, the application of generative models to virtual worlds, like financial markets, remains under-explored. In financial markets, generative models can simulate complex market effects of participants with various behaviors, enabling interaction under different market conditions, and training strategies without financial risk. This simulation relies on the finest structured data in financial market like orders thus building the finest realistic simulation. We propose Large Market Model (LMM), an order-level generative foundation model, for financial market simulation, akin to language modeling in the digital world. Our financial Market Simulation engine (MarS), powered by LMM, addresses the domain-specific need for realistic, interactive and controllable order generation. Key observations include LMM's strong scalability across data size and model complexity, and MarS's robust and practicable realism in controlled generation with market impact. We showcase MarS as a forecast tool, detection system, analysis platform, and agent training environment, thus demonstrating MarS's "paradigm shift" potential for a variety of financial applications. We release the code of MarS at `https://github.com/microsoft/MarS/`.

## 1 Introduction

The primary aim of generative models is to simulate realistic effects of various actions across different contexts, such as text generation (Achiam et al., 2023) and visual effects (Brooks et al., 2024). Real-world simulators enable human interaction with diverse scenes and objects (Mialon et al., 2023), allow robots to learn from simulated experiences without physical risk (Du et al., 2023), and generate vast amounts of realistic data for training other machine intelligence (Li et al., 2023).

While research on real-world simulators is extensive (Zhu et al., 2024; Yang et al., 2024), the application of generative models for virtual world simulation remains under-explored. The financial market exemplifies such a virtual world where each action, from trade execution to strategy deployment, can have ripple effects across a complex network of market participants. The ability to model and predict these effects in real time is crucial for traders, analysts, and regulators alike. Yet, current market simulation models – largely focused on statistical or agent-based approaches – lack the resolution, interactivity, and realism needed to reflect the full complexity of order-level behaviors.

To address these gaps, it is crucial to integrate the vast amounts of structured financial data, such as Limit Order Book (LOB) (Gould et al., 2013), that are essential for capturing market microstructures. We therefore propose the Large Market Model (LMM), a generative foundation model specifically designed for order-level financial market simulation. LMM builds on the successes of generative models in other domains but uniquely adapts them to the financial context, where the generation of orders, order batches, and LOBs plays a critical role in understanding market dynamics. By leveraging structured market data, LMM scales effectively with increasing data and model size, as we will demonstrate through scaling law evaluation, revealing its potential for handling large-scale

---

[*]Equal Contribution
[†]Corresponding Author

financial markets. LMM's design ensures that it can generate high-resolution market simulations, capturing both fine-grained individual order actions and broader market trends.

Powered by LMM, we introduce MarS, a financial Market Simulation engine, unlocking new potential in financial market forecasting, risk detection, strategy analysis. MarS is designed to ensure realism, producing simulated market trajectories that are robust enough for practical financial tasks such as predictive modeling, risk management, and agent training. It is capable of providing controlled generation, blending users' interactively injected orders into the generation of realistic market behaviors, assessing the market impact of these actions. This feature ensures that MarS delivers not only high-fidelity simulations but also controllable environments where financial strategies can be safely tested and evaluated.

Among the broad adoption of AI techniques in finance (Zhang et al., 2024; Liu et al., 2023b; Kim et al., 2019; Hou et al., 2021), MarS is the first to fully leverage the core elements of financial markets, making it a powerful tool for a wide range of downstream applications. We posit that MarS has the potential to bring paradigm shifts to a wide range of tasks related to the financial market. In this work, we demonstrate its transformative potential in four specific use cases:

1. **Forecast Tool**: MarS generates subsequent orders based on recent orders and LOB, simulating future market trajectories. This enables precise forecasting by analyzing multiple simulated trajectories.

2. **Detection System**: By generating multiple future market trajectories, MarS identifies potential risks not apparent from current observations. For example, a sudden drop in trajectory variance could indicate an impending significant event, providing early warnings and enhancing risk management.

3. **Analysis Platform**: MarS answers a wide range of "what if" questions by providing a realistic simulation environment. For instance, it evaluates the market impact of large orders by comparing existing market impact formulas to simulated results, identifying potential improvements and gaining deeper insights into market dynamics.

4. **Agent Training Environment**: The realistic and responsive nature of MarS makes it ideal for training reinforcement learning agents. This is demonstrated with an order execution scenario, showcasing MarS's potential for developing and refining trading strategies without real-world financial risks.

The main contributions of this paper are as follows:

- We introduce the Large Market Model (LMM), a generative foundation model designed specifically for financial market simulations, and demonstrate its scalability across data size and model complexity. This establishes a new direction for domain-specific foundation models in finance.

- We develop MarS, a high-fidelity financial market simulation engine powered by LMM, capable of generating realistic market scenarios and modeling the intricate impacts of order-level dynamics. This unlocks new possibilities for applying generative models in financial markets.

- We demonstrate the versatility of MarS through four key downstream applications: precise market forecasting, risk detection, market impact analysis, and agent training for trading strategies. These applications highlight the significant potential of MarS for transforming financial industry practices.

## 2 MarS Design

To create a truly realistic simulation system, MarS must excel in three key dimensions: high-resolution, controllability, and interactivity.

High-resolution refers to the ability of MarS to faithfully replicate the intricate dynamics of financial markets. This is why we leverage trading orders and order batches as the foundational elements of the simulation system, since they encapsulate the investment behaviors of market participants. These fine-grained data points are essential for accurately reproducing historical market trajectories, ensuring that the simulation reflects real market conditions and behaviors with precision.

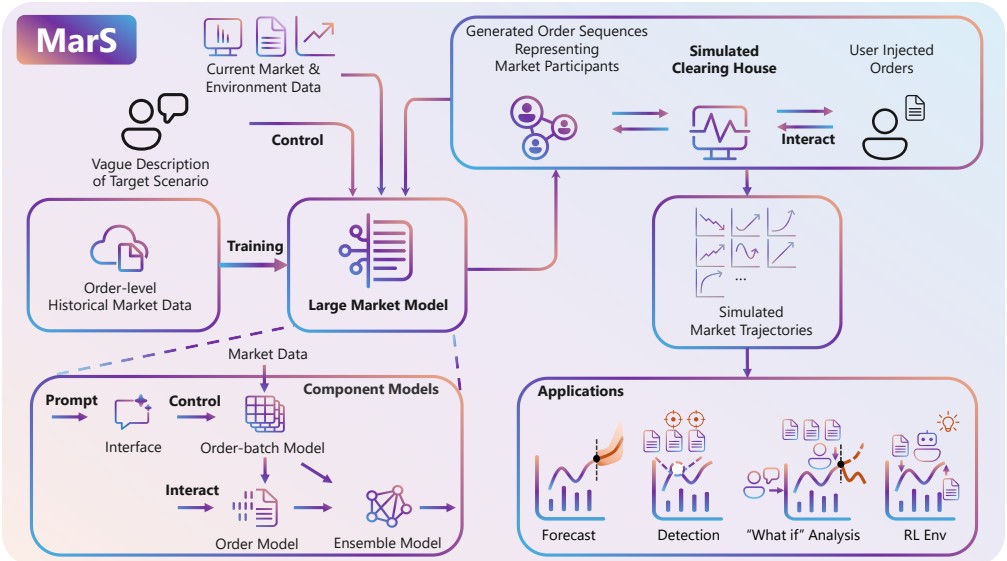

Figure 1: High-Level Overview of MarS. MarS is powered by a generative foundation model (LMM) trained on order-level historical financial market data. During real-time simulation, LMM dynamically generates order series in response to various conditions, including user-injected interactive orders, vague target scenario descriptions, and current/recent market data. These generated order series, combined with user interactive orders, are matched in a simulated clearing house in real-time, producing fine-grained simulated market trajectories. The flexibility of LMM's order generation enables MarS to support various downstream applications, such as forecasting, detection systems, analysis platforms, and agent training environments.

Controllability offers users the flexibility to simulate a wide range of market scenarios and circumstances. Under the scenarios of assessing market trends, monitoring potential risks, or optimizing trading strategies, MarS provides the tools needed to explore any possible market condition. This capability is particularly valuable for stress testing and strategy optimization, where diverse and even rare extreme cases must be modeled accurately.

Interactivity is crucial for enabling real-time user interaction with the simulated market. By allowing users to inject their own orders into the system, it enable them to evaluate market impacts, including both first-order and second-order effects. This feature is vital for analyzing trading strategies, managing systemic risks, and developing regulatory policies in a controlled, risk-free environment.

## 2.1 LARGE MARKET MODEL FOR FINANCIAL MARKET SIMULATION

**Problem Formulation.** To address the need for high-resolution, controllable, and interactive simulations, we propose the Large Market Model (LMM), a generative foundation model specifically designed for order-level financial market simulation. The problem is formulated as a conditional generation task, where the generation of trading orders is conditioned on historical data, user-injected orders, and market matching rules. LMM incorporates key features of the market microstructure such as Limit Order Books (LOB), enabling it to capture both individual trading behaviors and systemic market dynamics.

**Tokenization of Order and Order-Batch.** LMM models the generation of trading orders as a conditional generation process, leveraging sequential modeling techniques to predict the evolution of market states over time. This is achieved through a novel representation learning approach tailored for the financial industry's structured data, particularly the order flows at two distinct scales: individual orders and aggregated order-batches. The **Order Model**, using a causal transformer, tokenizes historical order sequences and Limit Order Book (LOB) information to ensure the realistic generation of individual trading orders. The tokenization procedure for the $i^{th}$ order is as follows:

$$Emb_i = \text{emb}(order_i) + \text{linear\_proj}(LOB_i^{\text{volumes}}) + \text{emb}(LOB_i^{\text{mid\_price}}), \tag{1}$$

where $order_i$ denotes an index indicating its position in the tuple (type, price, volume, interval), with type being one of ["Ask", "Bid", "Cancel"], $LOB_i^{\text{volumes}}$ represents the 10-level volumes for asks and bids in the LOB, and $LOB_i^{\text{mid-price}}$ is the mid-price of the LOB, expressed as the number of price tick changes since market opening.

In parallel, the **Order-Batch Model** converts the order batches into an image-like format, and employs VQ-VAE to represent and generate aggregated trading behaviors over discrete time intervals. In practice, we convert one order-batch into an RGB image format. We refer to such images as "order images", demonstrated in Fig. 2.

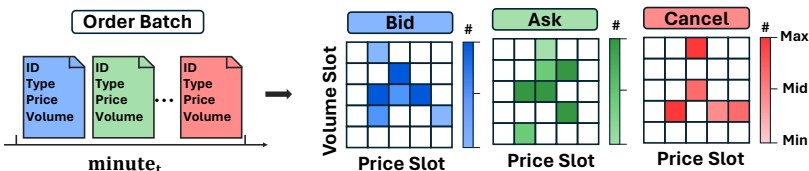

Figure 2: The order image converter transforms order data into a visual representation. Each order has three attributes: type (Bid, Ask, Cancel), price slot (relative to the mid-price), and volume slot (binned volume). The pixel values in the image represent the number of orders with the same attributes, with higher pixel values indicating more orders. More details can be found in C.3.

These components combine in an ensemble framework, where LMM uses auto-regressive modeling to build a foundational generative model. This framework integrates micro-level behaviors with macro-level market trends. LMM captures complex dependencies within historical data and temporal patterns through high-dimensional embeddings, providing robust market dynamics representation. For further details on the tokenization strategy and the architectural design of Order and Order-Batch Models, we refer the reader to Appendix B and C.

### 2.1.1 CONDITIONAL TRADING ORDER GENERATION

In LMM, the generation of trading orders is modeled as a conditional generation process that adapts to real-time market dynamics. An order clip is a sequence of trading orders $\mathbf{x} = (x_0, \ldots, x_n)$, generated based on the following four key conditions: **DES_TEXT**: A general description of the desired market scenario (e.g., "price bump" or "volatility crush"), ensuring controllability. **Interactive Orders**: $(\dot{x}_{i+1}, \ldots, \dot{x}_{i+j})$ are user-injected orders after the $i$-th generated order. If $j = 0$, there are no interactive orders between $x_i$ and $x_{i+1}$. **Starting Sequence**: $(x_0, \ldots, x_{m-1})$ are the initial $m$ orders, often using recent real orders to forecast subsequent ones, enabling realistic simulations. **MTCH_R**: Matching rules for trading orders, defining the feasible space for each order and reflecting the specific financial market's characteristics.

The conditional generation process: $p(x_{i+j+1}|\{DES\_TEXT, (\dot{x}_{i+1}, \ldots, \dot{x}_{i+j}), (x_0, \ldots, x_m), MTCH\_R\})$ ensures that generated orders are realistic and aligned with both the user-defined scenario and the underlying market structure. They can be adjusted for various MarS scenarios, with different applications showcased in Sec. 4. We provide a summary of the input conditions and configurations for various applications, along with the detailed introduction of **MTCH_R** and **DES_TEXT** in Appendix F.

### 2.1.2 FRAMEWORK DESIGN OF LARGE MARKET MODEL

The LMM integrates two complementary approaches: Order Sequence Modeling and Order-Batch Sequence Modeling, combined into an ensemble model to address financial market complexities. **Order Sequence Modeling.** We use a causal transformer to encode each order and its preceding Limit Order Book (LOB) information as a single token. This method captures the sequential nature of orders, ensuring realistic order sequences that reflect market dynamics. **Order-Batch Sequence Modeling.** To model structured patterns of dynamic market behavior over time intervals, we apply an auto-regressive transformer to order-batch sequences. Orders within a time step are grouped into batches, converted into a structured representation of market behavior for this time step, and modeled to maintain coherence and continuity. **Ensemble Model.** Combining order sequence and order-batch modeling, the ensemble model balances fine-grained control of individual orders with broader

market dynamics. This integration ensures detailed and contextually accurate market simulations. **Fine-grained Signal Generation Interface.** We introduce an interface that maps vague descriptions to fine-grained control signals using LLM-based historical market record retrieval. This guides the ensemble model, ensuring simulations follow realistic market patterns and user-defined scenarios.

The bottom-left of Fig. 1 shows the framework of the Large Market Model. The detailed design of its four parts can be found in Appendix B, C, D, E.

### 2.1.3 SCALING LAW IN LARGE MARKET MODEL

LMM's scalability is a key perspective to assess its effectiveness in handling increasingly large-scale financial markets. In our four-part foundation model design, we employ an auto-regressive transformer for order-batch sequences and a causal transformer for order sequences. These components utilize standard pre-training techniques commonly applied in foundation models, including those used in language modeling (Kaplan et al., 2020) and vision modeling (Zhai et al., 2022).

To assess the scalability of the LMM, we evaluated its performance across varying data scales and model sizes. The scaling curves are shown in Fig. 3. Our findings indicate that as the size of the data and the model increases, LMM's performance improves significantly, consistent with the scaling laws observed in other foundation models. This suggests that the potential of LMM can be further unlocked by leveraging larger datasets and more extensive computational resources.

While the current implementation only taps into a fraction of the available order-level financial market data due to resource constraints, the vast amount of data accessible within financial markets holds tremendous promise for future enhancements. MarS, in this context, serves as the tool to unearth this "gold mine" of data, indicating substantial opportunities for more comprehensive and powerful market simulations.

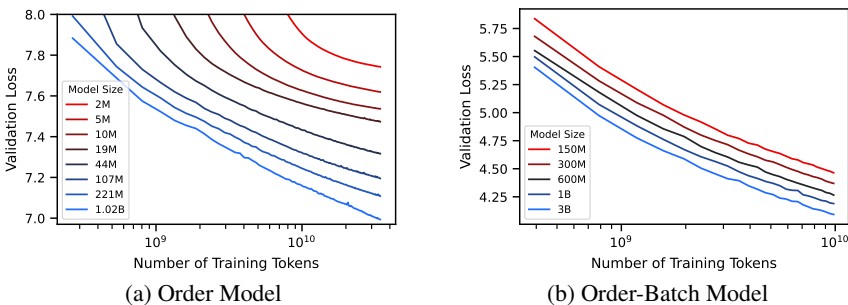

|        (a) Order Model        |        (b) Order-Batch Model        |

Figure 3: Scaling curves of Order Model and Order-Batch Model. (**a**) Order Model: Trained on 32 billion tokens, with model sizes ranging from 2 million to 1.02 billion parameters. (**b**) Order-Batch Model: Trained on 10 billion tokens, with model sizes ranging from 150 million to 3 billion parameters. The results demonstrate enhanced performance with increased data and model sizes.

### 2.2 MARS — ORDER GENERATION COMBINED WITH SIMULATED CLEARING HOUSE

Powered by LMM, the MarS engine is designed to generate highly realistic market trajectories that are robust enough for practical financial tasks such as predictive modeling, risk management, and agent training.

At the core of MarS is the simulated clearing house, which matches both generated and interactive orders in real-time, providing extensive information (e.g., LOB) for subsequent order generation. For each generated order $x_i$, the clearing house processes it against any $j$ interactive orders ($j \geq 0$) injected by the user. The results of this matching process, including the recent LOB, are then used to generate the next order $x_{i+j+1}$, creating a continuous and dynamic simulation.

MarS excels at providing controlled generation, blending users' interactively injected orders into the generation of realistic market behaviors. Users can inject their own orders into the system and observe how these actions impact market dynamics in real-time. This capability allows users to simulate various trading strategies, assess market impacts, and evaluate the performance of their strategies under different conditions. The blending process is carefully managed in MarS by adhering to two guiding principles.

- **"Shaping the Future Based on Realized Realities."** At each time step, the order-batch model generates the next order-batch based on recent orders and corresponding matching results from the simulated clearing house. These information conclude the immediate market impact of users' injected orders and determines the generated market behaviors in the next order-batch.

- **"Electing the Best from Every Possible Future."** Multiple predicted order-batches are generated at each time step and the best match to the fine-grained control signal is selected, ensuring the simulation remains realistic while allowing for user control.

The order-level transformer, trained on historical orders, naturally learns immediate market impact for subsequent order generation. Concurrently, the ensemble model influences order generation, aligning with the generated next order-batch. Fig. 4 illustrates the generation process, balancing injected orders' market impact and control signals to form a realistic simulation.

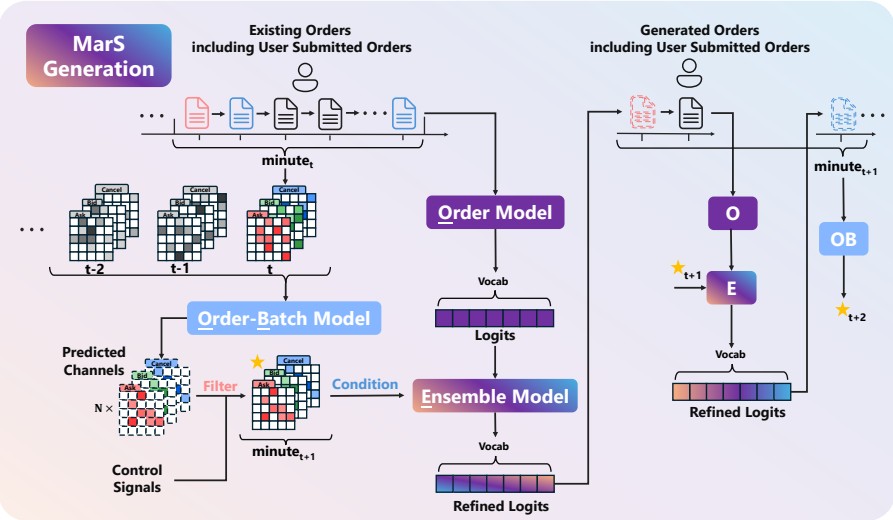

Figure 4: The process of MarS generation employs a two-level order generation mechanism. At the order-batch level, following the two guiding principles in Sec. 2.2, the Order-Batch Model processes existing orders from $minute_t$ and generates $N$ possible distributions for $minute_{t+1}$. Through a filter process based on control signals, the target distribution ($\star$) is selected and serves as a condition for the Ensemble Model (E). At the order level, the Order Model (O) generates immediate responses for recent and user-submitted orders, while the Ensemble Model refines these generations conditioned on the target distribution. The generated orders in $minute_{t+1}$ are fed back to the Order-Batch Model (OB) for $minute_{t+2}$ prediction, creating a dynamic feedback loop that balances market impact and controlled generation.

## 3 EXPERIMENTS

This section evaluates the capabilities of MarS in providing realistic, interactive, and controllable simulations. Note that throughout our experiments, the term "**replay**" refers to replaying real historical market data within MarS to validate the simulation against real-world events.

### 3.1 REALISTIC SIMULATIONS

To assess the realism of MarS's market simulations, we compare simulated data against key stylized facts derived from historical market data (Sherkar & Sen, 2023). These stylized facts serve as robust benchmarks, ensuring market simulations accurately reflect real-world market behaviors (Vyetrenko et al., 2020; Coletta et al., 2022; Stillman et al., 2023). Fig. 5 presents several prevalent stylized facts. MarS successfully replicates these stylized facts, demonstrating its capability to produce highly realistic market simulations suitable for practical applications. Besides these three stylized facts, we provide a detailed evaluation of other **eleven** stylized facts in Appendix I and a quantitative analysis in Appendix J.

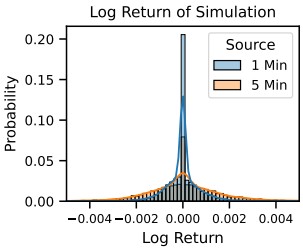 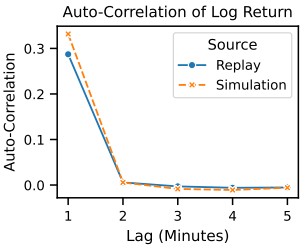 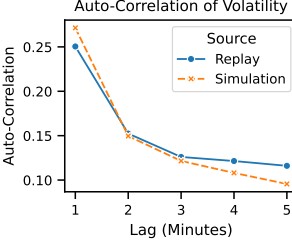

| (a) Aggregational Gaussianity | (b) Absence of Autocorrelations | (c) Volatility Clustering |
|---|---|---|

Figure 5: Illustration of Stylized Facts in MarS. (**a**) Aggregational Gaussianity: as the interval increases from 1 to 5 minutes, the distribution of log returns becomes more similar to a normal distribution. (**b**) Absence of Autocorrelations: the auto-correlation of log returns rapidly decreases with increasing intervals. (**c**) Volatility Clustering: high volatility auto-correlation is observed over periods.

### 3.2 INTERACTIVE SIMULATIONS

Understanding market impacts, i.e., changes in financial markets caused by trading activity, is crucial. MarS simulates these impacts by generating orders from detailed order-level data. Fig. 6a illustrates MarS interacting with a trading agent executing a TWAP (Time-Weighted Average Price) strategy, which caused observable changes in the subsequent price trajectory. The gap between the two curves represents the synthetic market impact generated by the agents trading actions. A detailed exploration of market impact can be found in Sec.4.3.

We validated these simulations by collecting market impacts from agents with various configurations, confirming that the synthetic data adheres to the *Square-Root-Law*, as depicted in Fig. 6b. The *Square-Root-Law*, $\Delta \propto \sigma \sqrt{Q/V}$, is a widely used model for market impact (Moro et al., 2009; Lillo et al., 2003; Almgren et al., 2005), where $\Delta$ is the price change, $\sigma$ is the volatility, $Q$ is the trading volume, and $V$ is the total market volume. These results illustrate that MarS can effectively model the impact of trading strategies on market prices, providing valuable insights for market participants and aiding in the development of more robust trading strategies. Additional details and results about the TWAP agent and market impact can be found in Appendix H and K.

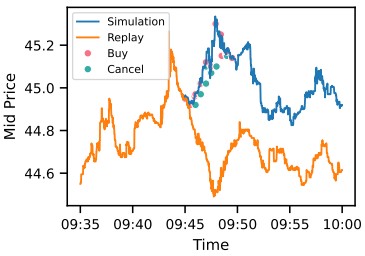 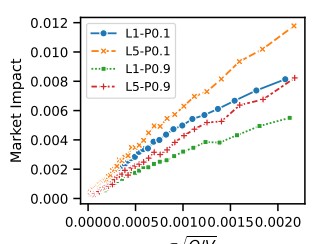 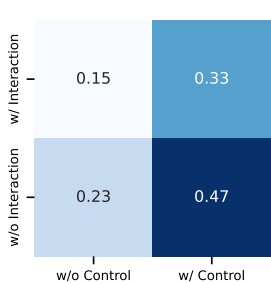

| (a) Synthetic market interaction | (b) Square-Root-Law Validation | (c) Effects of control signals |
|---|---|---|

Figure 6: Results of interactive and controllable simulations in MarS.

### 3.3 CONTROLLABLE SIMULATIONS

We demonstrate the controllability of MarS by replicating historical events. Specifically, MarS allows two types of control signals: {**replay curve**, **prompt**}. For control with **replay curve**, we simulate a price change between 0.3% and 0.5% over 5 minutes. With control enabled, an order batch is generated using minute-level guiding signals from the replay curve, integrated with the order model within an ensemble model to produce trading orders. Fig. 6c depicts the correlation between simulated and replay price trajectories. The introduction of control signals significantly enhances the correlation scores ($0.23 \rightarrow 0.47$), showcasing MarS's effectiveness in generating controllable market simulations. Fig. 6c shows the balance between control and interaction. Configurations with control but no interaction achieve the highest correlation scores, while introducing interaction reduces control precision ($0.47 \rightarrow 0.33$). This inherent balance allows for more realistic interactions in diverse applications. For control with **prompt**, MarS allows users to use natural language to describe specific historical scenarios, then utilizes Large Language Models(LLMs) to guide the generation through the **fine-grained signal generation interface**. The detailed results are provided in Appendix E.

# 4 APPLICATIONS

In Sec.2 and 3, we demonstrated the formulation of diverse financial tasks as a conditional trading order generation problem. Our experiments showed that MarS is Realistic, Controllable, and Interactive, establishing it as a robust financial market simulator. This section explores potential downstream applications of MarS, further validating its foundational role in financial market simulation. We present practical financial tasks to illustrate: a) MarS's capability to solve financial problems independently, and b) its utility as a simulation platform for other tasks. For a), we showcase Forecast and Detection tasks, and for b), we provide examples of "What if" Analysis, and Reinforcement Learning Environment.

Here, we highlight that, analogous to text generation vs. language modeling (Achiam et al., 2023; Abdin et al., 2024; Dubey et al., 2024), and video generation vs. physical world decision making (Liu et al., 2024; Yang et al., 2024; 2023a), we have constructed a unified task interface through conditional trading order generation for diverse financial downstream tasks with MarS. This interface can transfer complex and diverse financial information into specific tasks. We compare current methodologies with the new paradigm introduced by MarS to illustrate the "paradigm shift" across various types of financial tasks, as shown in Table 1. Detailed introductions are provided in the subsequent sections.

| Applications | Current Methods | MarS |
|:---:|:---:|:---:|
| Forecasting | sequence extrapolation | conditional generation |
| Detection | $\text{Diff}(market_{now}, market_{past})$ | $\text{Diff}(market_{now}, simu\text{-}market_{now})$ |
| "What if" Analysis | online experiments, empirical formula | offline data-driven pipeline |
| RL Environment | finite data, fake $P(s_{t+1}|s_t, a_t)$ | infinite data, real $P(s_{t+1}|s_t, a_t)$ |

Table 1: Summary of how MarS reshapes mainstream financial applications. $\text{Diff}(\cdot, \cdot)$ represents the difference between two market states for anomaly detection. $P(s_{t+1}|s_t, a_t)$ denotes the state transition given the current state and action. Without an interactive environment, most existing financial RL works cannot model the realistic impact of market state caused by agent actions. Further details of the RL-Environment are in Sec.4.4.

## 4.1 FORECASTING

Forecasting is crucial in many financial applications, with market trend forecasting being a prime example. This task demands models that accurately capture and reflect market dynamics. Traditionally, direct forecasting models are used. In this section, we assess the effectiveness of our market simulation in predicting trends.

Following Ntakaris et al. (2018), we define the price change from $t$ to $t + k$ minute as: $l = \left(\left(\frac{1}{n}\sum_{i=1}^{n} m_i\right) - m_0\right)/m_0$, where $m_0$ is the mid-price at time $t$, $n$ is the number of orders between $t$ and $t + k$ minutes, and $m_i$ is the mid-price after the $i$th order event. The price change is categorized into three classes—up, down, and flat—based on the value of $l$, ensuring similar probabilities for each class over the training period. We compare our model with DeepLOB by Zhang et al. (2019), a well-known baseline. Fig. 7a illustrates that LMM-based simulations significantly outperform DeepLOB, highlighting its superior market dynamics understanding. Additionally, the 1.02 billion-parameter model outperforms the 0.22 billion-parameter model, indicating that improved validation loss in scaling curve (Fig. 3) correlates with enhanced forecasting performance.

It is noteworthy that all forecasting targets can be calculated using simulated trajectories from MarS, whereas traditional direct forecasting models require separate training for each target. This underscores the significant advantage of simulation-based forecasting by MarS. For more discussion about the comparison between DeepLOB and MarS/LMM, please refer to Appendix L.

## 4.2 DETECTION

Detecting the changing state of market is crucial in financial tasks, especially in the regulation of market abuse, e.g., insider trading (Meulbroek, 1992) and market manipulation (Putniņš, 2012). We demonstrate how MarS could bring a new simulation-based paradigm to detection tasks by monitoring the similarity between simulated and real market patterns. Using real market manipulation cases from CSRC[1], we evaluate the similarity of spread distributions through Distribution Similar-

---

[1]http://www.csrc.gov.cn

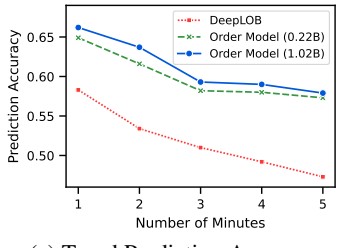 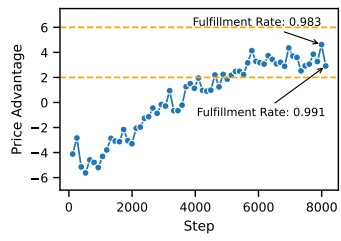

(a) Trend Prediction Accuracy    (b) Performance of the trading agent

Figure 7: Results of forecasting and RL-agent training tasks. For forecasting task, MarS executes 128 simulations at each initial time point, and aggregate outcomes to determine the final predicted class. The ground truth is obtained from historical replay. For RL-agent training, the x-axis represents the number of update batches, and the y-axis is the price advantage over our best-configured TWAP agent (L1-P0.9), in basis points (BP).

ity[2], which serves as a key indicator of market liquidity. While MarS maintains high distribution similarity ($> 0.87$) in normal periods, its simulation realism drops significantly during manipulation periods, particularly showing a heavier tail and a peak around $\delta = 1000$ (Fig. 8). These anomalies can be viewed as signals likely corresponding to market manipulation, where manipulators significantly impact liquidity. This suggests a promising direction for automated anomaly detection, though comprehensive evaluation combining multiple metrics is necessary for robust conclusions. Detailed analysis and experimental settings are provided in Appendix G.

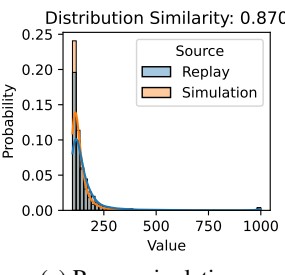 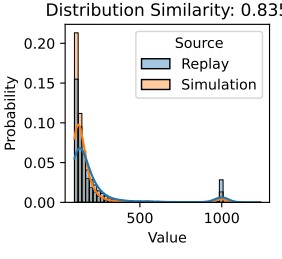 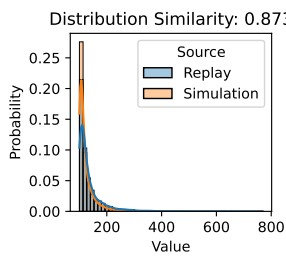

(a) Pre-manipulation    (b) Manipulation period    (c) Post-manipulation

Figure 8: Spread distribution in different periods of market manipulation. The distribution similarity between replay and simulation drops during the manipulation period (b), where a heavier tail and a noticeable peak around $\delta = 1000$ emerge, in contrast to the pre-manipulation (a) and post-manipulation (c) periods.

### 4.3 "WHAT IF" ANALYSIS ON MARKET IMPACT

One of the most important "What if" topics in finance is to analyze market impact, the change in asset prices caused by trading activity. Due to complex mechanisms, most existing research in this area relies heavily on strong assumptions and empirical formulas (Zarinelli et al., 2015; Almgren et al., 2005; Gatheral, 2010; Gatheral et al., 2012; 2011), and is limited to costly and risky online experiments. In this section, we take market impact as an example, showing how MarS can act as a reliable and powerful platform and contribute to "what if" analysis. As we have validated the reliability of synthetic market impact in Sec.3.2, we step to a more ambitious goal: leverage the synthetic data to build data-driven pipeline to discover new laws to explain market impact and its long-term dynamics. Due to the limited space, details of experiment settings, clarification, and more results in this section are provided in Appendix K.

**New factors beyond Square-Root-Law:** To uncover new factors beyond Square-Root-Law influencing market impact, we first employed symbolic regression (de Silva et al., 2020), using classic volume and price factors before trading as the base dictionary. By applying genetic algorithms, we sought to identify the most informative factors on synthetic market impact. The preliminary results were reviewed and refined by domain experts, leading to the discovery of three new factors that partially explain market impact: {*resiliency*, *LOB_pressure*, *LOB_depth*}. We show the relationship between market impact and factor *resiliency* in Fig. 9a.

**Dynamics of Long-Term Market Impact:** The long-term market impact, also known as price impact trajectory, typically manifests as a gradually decaying sequence of price fluctuations after a trade. Traditional research relies on empirical formulas to model this dynamics (Gatheral et al.,

---

[2]https://en.wikipedia.org/wiki/Overlap_coefficient

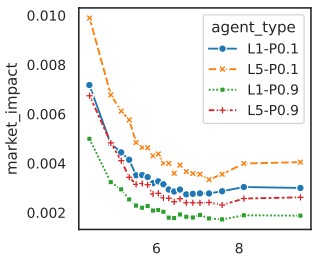
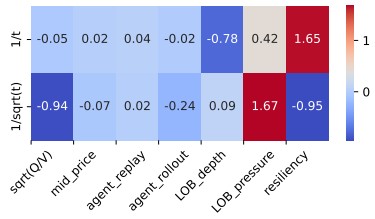

(a) New factor: Resiliency  (b) Interaction weights of learned ODE

Figure 9: Analysis of new market impact factor and long-term market impact.

2011; Donier et al., 2015a; Bacry et al., 2015), but could struggle to capture the full complexity of real-world scenarios. To address this, we leverage generated market impact to develop a more accurate, data-driven approach. Our method models the decay dynamics using an ordinary differential equation (ODE), which integrates both potential influencing factors and decay functions:

$$\frac{dY(t)}{dt} = \text{sum}(W \circ (X \otimes F^{\text{decay}}(t))) = \sum_{i=1}^{m} \sum_{j=1}^{n} W_{i,j} X_i F_j^{\text{decay}}(t) \qquad (2)$$

where $Y(t)$ is the long-term market impact, $X \in \mathbb{R}^m$ is the factor group, such as volume, price, etc., and $F^{\text{decay}}(t) : t \to \mathbb{R}^n$ includes possible decay functions, e.g., $[1/t, \ldots, 1/\sqrt{t}]$. $X$ and $F^{\text{decay}}(t)$ can be customized based on domain knowledge. $\otimes$ is the outer product, $\circ$ is the Hadamard product, $X^T \otimes F^{\text{decay}}(t)$ is a matrix with size $\mathbb{R}^{n \times m}$, representing interactions among factors and decay patterns, and $W \in \mathbb{R}^{n \times m}$ is the learnable interaction weight. Fig. 9b shows the learned weights $W$, demonstrating the importance of interaction pairs of two decay functions and seven factors, which can help to deepen our understanding of the long-term market impact.

### 4.4  REINFORCEMENT LEARNING ENVIRONMENT

The MarS environment, being both realistic and interactive, is ideal for training reinforcement learning (RL) agents. This environment accurately reflects an agent's impact, provides realistic rewards, and facilitates training robust agents for the financial market. In this experiment, we aim to train a trading agent from scratch using MarS. The trading agent's goal is to purchase a large volume within 5 minutes, optimizing both fulfillment rate and price advantage.

The trading agent's state includes features such as remaining time, remaining volume, LOB imbalance, and the period's stage (passive or aggressive). The agent's actions are based on a configurable TWAP strategy and the reward function is defined as follows:

$$\text{Reward} = \alpha \times \text{FulfillmentRate} + \text{PriceAdvantage}, \qquad (3)$$

where $\alpha = 1$ when $\text{FulfillmentRate} \leq 0.95$ and decreases to 0 as FulfillmentRate approaches 1. Detailed settings of agent training can be found in Appendix H.

Fig. 7b shows the training performance of the trading agent. The agent's performance improves from -6 BP to 2~6 BP during training. The observed fluctuations between 2~6 BP are attributed to the agent exploring various strategies between high and low fulfillment rates, resulting in corresponding variations in price advantage based on the current reward setting. This demonstration highlights that MarS is capable of training trading agents from scratch by leveraging its realistic and interactive simulation capabilities.

## 5  RELATED WORK

We give a detailed and comprehensive discussion of related work on financial market simulation and generative foundation models in Appendix A.

## 6  CONCLUSION

We introduce MarS, an order-level, fine-grained realistic financial market simulation engine, powered by the generative foundation model, LMM. Our evaluation of LMM's scaling law demonstrates the potential for continuous improvement in future financial world models. We identify three essential characteristics of impactful market simulation: realism, controllability, and interactivity. We present four representative tasks developed using MarS, underscoring its potential to catalyze a paradigm shift across various financial applications.

ACKNOWLEDGEMENTS

We would like to thank our colleagues Xiao Yang and Xu Yang for contributing to our early prototype and their invaluable feedback and suggestions during our regular discussions. We also express our sincere gratitude to Chengqi Dong for his meticulous analysis and exploration on market impact data, which have significantly enhanced our understanding of market impact studies.

DISCLAIMER

Users of the market simulation engine and the code should prepare their own agents which may be included trained models built with users own data, independently assess and test the risks of the model in a specify use scenario, ensure the responsible use of AI technology, including but limited to developing and integrating risk mitigation measures, and comply with all applicable laws and regulations. The market simulation engine does not provide financial opinions, nor is it designed to replace the role of qualified financial professionals in formulating, assessing, and approving finance products. The outputs of the market simulation engine do not reflect the opinions of Microsoft.

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

## A   RELATED WORKS

**Financial Market Simulation.** Before the recent surge in generative foundation models, researchers in the finance domain had already recognized the immense potential of powerful market simulations. Early approaches often utilized agent-based modeling, particularly multi-agent systems, to simulate order-driven markets (Chiarella et al., 2009; Byrd et al., 2020; Amrouni et al., 2021).

With the advancements in deep learning technologies, several works have emerged that adopt the world model paradigm to simulate Limit Order Book (LOB) markets (Takahashi et al., 2019b; Li et al., 2020; Coletta et al., 2021; 2022; 2023). These studies primarily leveraged Generative Adversarial Networks (GANs) (Goodfellow et al., 2020) to model the distribution of LOB time series.

Recently, some generators have begun incorporating market micro-structure data, such as those presented in (Hultin et al., 2023; Nagy et al., 2023). Among these, Nagy et al. (2023) is most related to our work, particularly regarding the order model. They employ an auto-regressive model based on a Deep State Space Network (Rangapuram et al., 2018) to generate LOB and message flows. However, their focus is primarily on LOB modeling. While they demonstrate some realistic stylized facts of the generated sequences, they do not evaluate the model's capability to address downstream financial tasks.

Our work aims to push the boundaries of financial market simulation by introducing an innovative approach that goes beyond generating realistic order flows. We introduce MarS, a pioneering financial market simulation engine driven by the Large Market Model (LMM). Designed to meet the specific demands of the financial sector, MarS excels in modeling the market impact of orders and achieving high levels of controllability and realism. By framing various financial market tasks as conditional trading order generation problems, we demonstrate MarS's transformative potential and practical applications in real-world financial markets.

**Foundation Models.** Foundation models are trained on broad datasets and can be adapted to a wide range of downstream tasks. The term was popularized by the Stanford Institute (Bommasani et al., 2021). The release of GPT-3 (Brown, 2020) showcased the powerful benefits of training large auto-regressive language models (LLMs) on extensive corpora (Abdin et al., 2024; Achiam et al., 2023; Dubey et al., 2024).

In addition, numerous foundation models have emerged in the fields of computer vision (CV) and multimodal areas (Rombach et al., 2021; Brooks et al., 2024; Liu et al., 2023a). Recently, real-world simulators and industry-specific large models have become popular research topics in this field. Real-world simulators aim to achieve real-world simulation through the unified goal of video generation, addressing various tasks in fields such as autonomous driving, robotics, and gaming (Liu et al., 2024; Zhu et al., 2024; Yang et al., 2024; 2023a). However, they primarily focus on simulating the physical world. The order-driven financial market is an exemplary virtual world with different operating principles. To the best of our knowledge, we are the first to build a financial world simulator.

Industry-specific large models primarily focus on fields such as biomedicine (Moor et al., 2023), law (Huang et al., 2023), and finance. In the financial domain, most large models are Financial LLMs, which either pre-train LLMs on financial corpora (Wu et al., 2023; Zhang & Yang, 2023) or fine-tune them (Xie et al., 2024a; Zhang et al., 2023; William Todt, 2023; Yang et al., 2023b) to tackle financial NLP tasks or multimodal tasks (Bhatia et al., 2024; Xie et al., 2024b), including sentiment analysis, text classification, and question answering.

Beyond text, there is an even larger and more information-rich corpus in the financial world: trading orders. We propose a Large Market Model (LMM), which, for the first time, reveals the scaling law on trading orders. We take the first step toward building a generative foundation model as a world model for the financial market. We believe that, with MarS as the shovel, the extensive order-level data undoubtedly represent a significant gold mine.

# B ORDER SEQUENCE MODELING.

## B.1 INTRODUCTION

The order model for financial markets shares similarities with the Language Model (LM) for text in several respects. Both models strive to predict the subsequent event, whether it be a token in a text corpus or a trade order in financial markets. Additionally, the datasets for both are typically extensive, facilitating the training of robust models. Furthermore, data in both domains can be generated autoregressively.

Nevertheless, substantial differences also exist between the two fields. Each order in the financial market is associated with a complex set of market dynamics, including the Limit Order Book (LOB), transactions, and potentially market news in natural language. Consequently, each order may be influenced by a broader array of information beyond the order stream itself. It is therefore imperative to encode this rich information compactly while preserving the autoregressive generation paradigm. Moreover, the financial market operates on a rule-based order matching system, which processes orders and generates new states, such as transactions and the updated LOB. This necessitates an additional order matching step to obtain accurate market states.

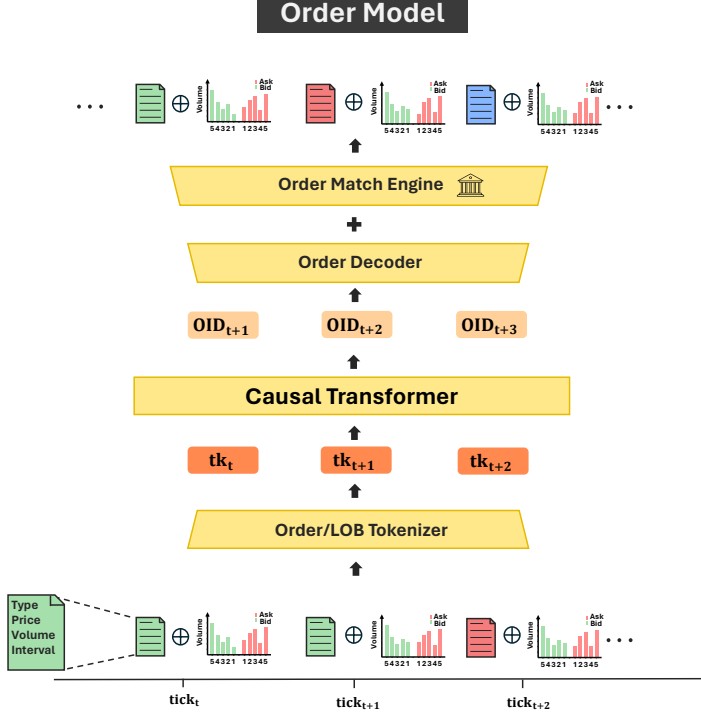

Figure 10: The framework of the order model. The model is trained on the order stream and the corresponding LOB information. It is autoregressive, generating the next order based on the preceding order and LOB information. The order matching step is employed to produce the new LOB state.

## B.2 APPROACH

### B.2.1 TOKENIZATION

The objective of tokenization is to make it compact and efficient for encoding and decoding while retaining the majority of useful information. To this end, we opt to encode each order and its antecedent LOB as a single token. The LOB information functions analogously to an image in a text, offering additional context for the order. The tokenization procedure for the $i^{th}$ order is as follows:

$$Emb_i = \text{emb}(order_i) + \text{linear\_proj}(LOB_i^{\text{volumes}}) + \text{emb}(LOB_i^{\text{mid-price}}) \tag{4}$$

Here, $order_i$ denotes an index indicating its position in the tuple (type, price, volume, interval), with type being one of ["Ask", "Bid", "Cancel"]. Both price and volume are discretized into the range [0, 32), and interval into [0, 16). An index within the range [0, 49152) can uniquely identify a position for the (type, price, volume, interval) tuple. $LOB_i^{\text{volumes}}$ represents the 10-level volumes for asks and bids in the LOB, also discretized into [0, 32). The $LOB_i^{\text{mid\_price}}$ is the mid-price of the LOB, expressed as the number of price tick changes since market opening.

This formula computes the embedding for the $i^{th}$ token, which is a composite of the order, the linear projection of the LOB volumes, and the embedding of the LOB mid-price.

While the input token includes LOB information, it is impractical and unnecessary to predict the resultant LOB during the decoding process. Instead, the new LOB information can be derived using a standard order matching algorithm, based on the preceding LOB and the newly generated order. Given this consideration, we only output the order index and conduct an order matching during simulation to obtain the subsequent accurate LOB state, as depicted in Fig. 10.

### B.3    DATA AND MODEL TRAINING

Our dataset encompasses the top 500 liquidity stocks in the Chinese stock market, covering the period from 2017 to 2023 and comprising 16 billion order tokens. Our model architecture is based on LLaMA2 (Touvron et al., 2023), and AdamW optimizer (Loshchilov, 2017) is employed in all experiments. We utilize fp16 precision with DeepSpeed ZERO stage 2 (Rajbhandari et al., 2020) to optimize memory usage. The sequence length is set at 1024, with a batch size of 4096, equating to 4 million tokens per optimization step.

The inclusion of LOB information in the tokenization process is compared to determine its impact on training performance. The evidence suggests that integrating the LOB information contributes to an enhanced training curve, as shown in Fig. 11.

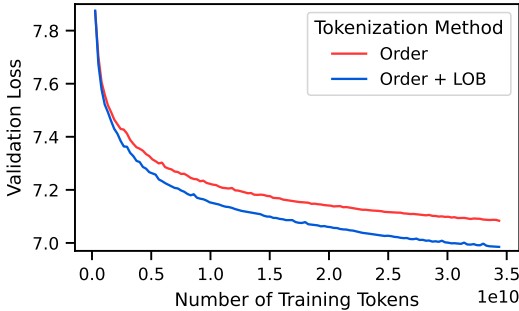

Figure 11: Tokenization of the Order Model. A comparative analysis of the tokenization process with and without the Limit Order Book (LOB) information. Incorporating precise LOB information leads to an improved training curve.

Furthermore, we examine the effects of varying data and model sizes on training performance. The data suggest that augmenting both data and model sizes correlates with improved outcomes, as shown in Fig. 3a.

## C    ORDER-BATCH SEQUENCE MODELING

### C.1    INTRODUCTION

In this section, we introduce the order-batch model. Different from the order model, which focuses on individual orders, the order-batch model concentrates on batches of orders to model structured patterns of dynamic market behavior over time intervals. We innovatively organize batches of orders into an RGB image format, which are then discretized into tokens for autoregressive training, aimed at generating order-batch sequences.

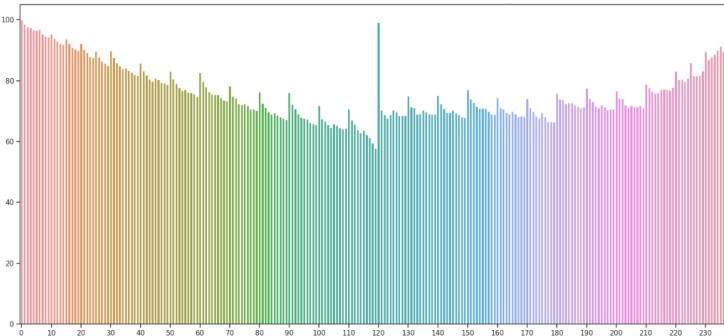

Figure 12: The intraday distribution of the average number of orders per minute.

As we know, financial markets are comprised of diverse participants, each with a unique set of information and trading frequency. Even in the domain of high-frequency trading, there are nuances: some traders pay close attention to each order, while others may focus on signals in fixed time intervals to guide their trading decisions. Through data analysis, we can easily discern the traces by the latter type of high-frequency traders. We counted the number of orders per minute for each stock in our dataset introduced in Sec. B.3 to create a chart shown in Fig. 12. From this chart, we can observe the following patterns: 1. The intraday order distribution is U-shaped. 2. There is a significant increase in order number at the market open in the morning and after the lunch break. 3. There are spikes in order numbers nearly every 10 minutes, suggesting a periodic pattern. With the above observations, we find that the distribution of orders within fixed intervals adheres to consistent patterns, and such patterns can also be captured by the model. So we attempt to model these structured patterns of dynamic market behavior.

Besides, modeling batches of orders facilitates the generation of specific financial scenarios. If generating a specified market scenario through prompts, there will be significant information asymmetry between the brief text of prompts and the thousands of orders in an order flow. Imposing such a signal directly onto each order through an order model is clearly intractable. Therefore, we need an order-batch model to act as a bridge between the prompt and the order model to facilitate this transition. The order-batch model corresponds to prompts by first generating minute-level order batches, and then decoding them into an order flow in conjunction with the order model.

### C.2   APPROACH

As observed in Fig. 12, orders within fixed time intervals vary in numbers, and these variations are significant at different time throughout the day. In light of this, learning representations from the sequences after padding is clearly not a sensible approach. To better represent orders of variable numbers, we creatively convert the orders into an RGB image format. This approach allows us not only to "visualize" the changes in orders over a period of time but also to draw on the experience of the image generation field, transforming the problem of order-batch generation into one of image generation. We present the framework of the order-batch model in Fig. 13.

### C.3   ORDER IMAGE CONVERTER

Learning representations directly from order sequences at fixed time intervals is not an effective and practical approach. On the one hand, stocks with different levels of liquidity have significantly different order numbers. On the other hand, for the same stock, the distribution of order numbers throughout the day can be extremely uneven (with a higher concentration during the opening and closing periods, and sparser distribution during the mid-day). Within fixed time intervals (e.g., minute-level), we care more about the aggregate characteristics of the order sequence rather than the details of individual orders. Under the assumption that the distribution of orders remains relatively stable over short periods, we can disregard the precise arrival times of individual orders and structure the order sequences in a cross-sectional view.

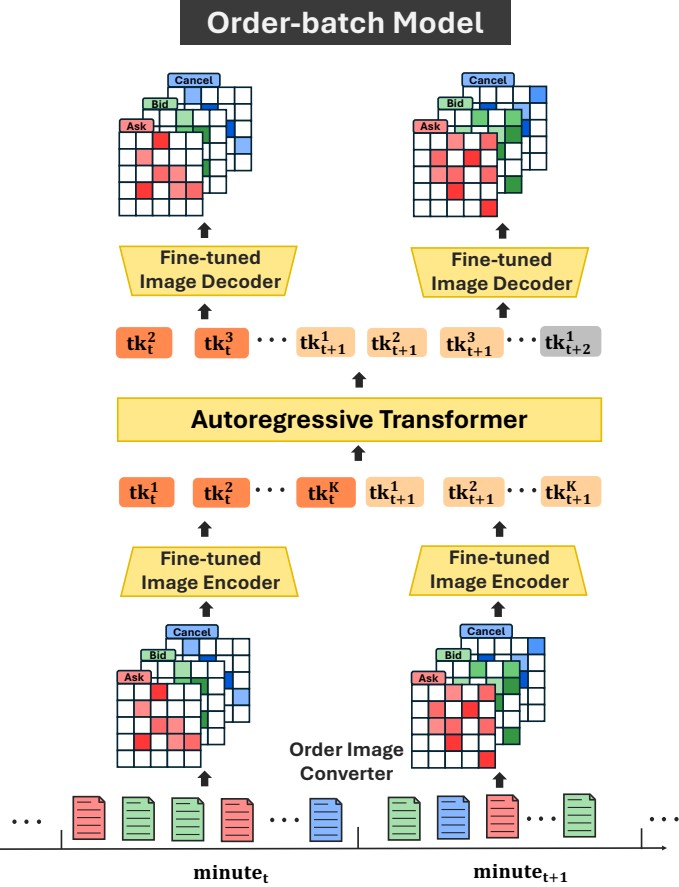

Figure 13: The framework of order-batch model. We employ a two-stage training approach: in Stage 1, we leverage a fine-tuned image encoder to transform "order images" from minute-level orders into tokens; in Stage 2, we train an autoregressive transformer model to learn the distribution of the tokens. Order images are decoded from tokens via fine-tuned image decoder.

In practice, we convert one order-batch into an RGB image format. We refer to such images as "order images" with shape $[C, H, W]$, as we demonstrated in Fig. 2. $C$ denotes the categories of orders, or the channels of an order image. $W$ and $H$ represent the width and height of the order image, indicating the number of price and volume slots, respectively. The pixel value $V$ of the order images signifies the count of identical orders. In our work, we set $C = 3, H = W = 32, V \in [0, 100]$.

The order image converter allows us not only to "depict" the changes in orders over a past period but also to leverage experience from image generation. We can utilize a pre-trained visual encoder to obtain an order-batch embedding.

### C.3.1    STAGE 1: ORDER IMAGE TOKENIZER

After converting the order-batch into an order image, we transform the problem of modelling order-batches into an image generation problem. In this way, we can follow the successful path of Large Vision Models (Bai et al., 2024), adopting a two-stage approach to generate intraday order-batch sequences. The first stage of the image generation task typically involves using a pre-trained image tokenizer to discretize individual images into a series of tokens.

Specifically, we leverage VQGAN (Esser et al., 2021) to accomplish the conversion of order images into discrete tokens, which learns a convolutional model consisting of an encoder and decoder, allowing them to represent images using codes from a learned, discrete codebook. In particular, VQ-GAN incorporates a discriminator and perceptual loss to ensure high quality during the compression

process. In our implementation, both the encoder and decoder utilized the original structure. **Technical Details**: We use a pre-trained VQGAN from LDM (Rombach et al., 2022), which was trained on the LAION-400M database (Schuhmann et al., 2021). We adopt the configuration and weights from one of the models in the LDM model zoo, with a down-sampling factor $f = 4$, vocabulary size $Z = 8192$, and codebook dimension $d = 3$. This means that an RGB order image of size $32 \times 32$ with 3 channels is discretized into $8 \times 8 = 64$ tokens at this stage, each with a dimension of 3. In practice, we find that the off-the-shelf model parameters did not represent order images well, so we fine-tune it using order images to achieve a transition from natural images to order images.

### C.3.2 STAGE 2: ORDER-BATCH SEQUENCE MODELLING

After the order image tokenizer converts individual order images into a sequence of discrete tokens, we concatenate these tokens to form an order-batch sequence. In Stage 2, we train an autoregressive transformer to learn the distribution of these tokens. It learns not only the distribution of tokens that make up an order image but also the distribution of tokens between order images. Consequently, we can generate intraday order-batch sequences.

Specifically, we employ a language model for next token prediction training. **Technical Details**: We use LLaMA2(Touvron et al., 2023) as the implementation framework for our autoregressive transformer. We calculate the cross-entropy loss between prediction logits and labels. Implementation Details: The token length for LLaMA2 is 4096, and we concatenate 16 order-batches to form an order-batch sequence, with a total length of $16 \times 64 = 1024$, which is well below the length limit.

## D ENSEMBLE MODEL

### D.1 INTRODUCTION

In sections above, we introduced the order model and order batch model, each with its advantages:

- **Order model**: This model generates orders individually and is designed to reflect short-term market impacts rapidly. However, it lacks the ability to generate target scenarios over the long run.
- **Order-batch model**: This model generates order channels (We do not distinguish 'order channels' and 'order images' in this paper), representing the macro behavior of the market, and can be used to follow control signals. However, it lacks the ability for interactive market simulation.

In this section, we introduce the ensemble model, which aims to balance interaction and controllability in market simulation.

### D.2 APPROACH

The order channels output by the order-batch model contain rich information about macro trends in the financial market. It would be advantageous if the order model could utilize this information to generate orders.

We propose using an ensemble model that takes the order logits and order channels as input and generates the next order, as illustrated in Fig. 4

In our experiment, we found it challenging to train the ensemble model directly from order channels predicted by the order-batch model. The reason is that the order channels predicted by the order-batch model still exhibit high variance and may not accurately reflect replay order data. Realizing this, during training, we use the order channels directly from replay data, which provides an accurate description of the market trend. In this way, our ensemble model learns how to condition on the order channels to generate the next order. During simulation, we use order channels predicted by the order-batch model to generate orders, which provide more flexibility for controllable simulation.

The ensemble model is a simple cross-attention model that takes the order logits and real order channels as input and generates the next order. The loss advantage over the order model is used as the training metric. Fig. 14 shows the training process of the ensemble model. We can see that with

this design, the ensemble model can improve its performance on order generation, demonstrating its conditioning on order batch data.

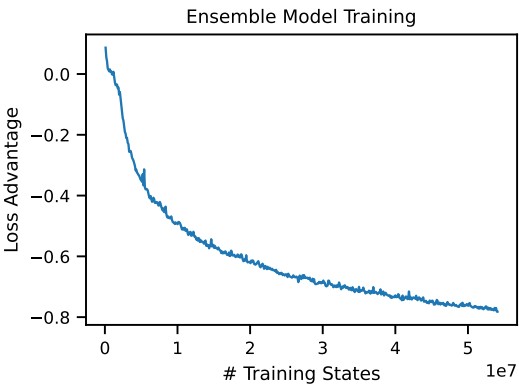

Figure 14: Training process of the ensemble model. The x-axis represents the number of training samples, and the y-axis represents the loss advantage over the order model.

## E  FINE-GRAINED SIGNAL GENERATION INTERFACE

We introduce an interface that maps vague descriptions to fine-grained control signals using LLM-based historical market record retrieval. This guides our order batch model, ensuring simulations reflect realistic market patterns and user-defined scenarios. The process involves three main steps:

- **Example Provision and Code Generation**: Provide a sample of minute-level return history to GPT-4o mini and prompt it to generate code that retrieves historical periods matching specified scenarios.
- **Scenario Filtering**: Apply the generated code on the entire dataset to identify more minute-level trajectories for each scenario.
- **Scenario Generation**: Use the identified minute-level trajectories to guide the generation of order batches according to principles outlined in Sec. 2.2, alongside the ensemble model for scenario generation.

The minute-level return history is stored in a CSV file, formatted as shown in Table 2:

| date | minute | SZ000001 | SZ000002 | ... | SZ003043 | SZ003816 |
|------|--------|----------|----------|-----|----------|----------|
| 2023-01-03 | 09:31:00 | -0.001520 | 0.001664 | ... | -0.005541 | 0.000000 |
| 2023-01-03 | 09:32:00 | 0.000761 | 0.000000 | ... | -0.004261 | 0.000000 |
| ... | ... | ... | ... | ... | ... | ... |
| 2023-03-31 | 14:55:00 | 0.000797 | 0.000657 | ... | 0.000164 | 0.000000 |
| 2023-03-31 | 14:56:00 | 0.000000 | -0.000656 | ... | 0.000164 | 0.000000 |

Table 2: Format of minute-level return history

We demonstrate market simulations for scenarios including "Sharp Drop", "Sharp Rise", and "Trend Reversal". Below, we detail the process when TEXT_DES is "Sharp Drop". First, a prompt is provided to GPT-4o mini, which generates code to filter typical cases for the "Sharp Drop" scenario. The prompt is shown in Table 3.

The code generated by GPT-4o mini, shown in Fig. 15, is then used to filter the "Sharp Drop" scenario and applied to the entire dataset to identify additional cases.

Once the minute-level return trajectory is retrieved, it is used to guide the generation of order batches along with the ensemble model for scenario generation. Detailed descriptions and visualizations of the three scenarios are provided:

```python
import pandas as pd
import datetime

file_path = 'minute_return_data_all.csv'
data = pd.read_csv(file_path)

data['datetime'] = pd.to_datetime(data['date'] + ' ' + data['minute'])
data.set_index('datetime', inplace=True)
data.drop(columns=['date', 'minute'], inplace=True)

# Apply groupby on each stock code to calculate rolling 25-minute sum
rolling_returns = data.groupby(data.index.date).rolling(25).sum().
                                    reset_index(level=0, drop=True)

# Filter based on sharp drops within the 25-minute windows (arbitrary
                                    threshold for "sharp drop")
threshold = -0.05   # Example threshold for a sharp drop over 25 minutes
sample_nums = 30
sharp_drops = rolling_returns[rolling_returns <= threshold].dropna(how='
                                    all')

# Reset the index to retain datetime information
sharp_drops = sharp_drops.reset_index()

# Extract 30 samples ensuring unique stock codes and trading dates, with
                                    varied start times
result = []
seen_dates = set()
seen_stocks = set()
seen_start_times = set()

for _, row in sharp_drops.iterrows():
    date = row['datetime'].date()
    start_time = (row['datetime'] - pd.Timedelta(minutes=24)).time()
    if start_time <= datetime.time(9, 30) or start_time <= datetime.time(
                                        13, 00) and start_time >
                                        datetime.time(11, 30):
        continue
    stock_drops = row.drop(labels=['datetime'])

    if start_time not in seen_start_times:
        for stock_code, value in stock_drops.items():
            if not pd.isna(value) and date not in seen_dates and
                                        stock_code not in
                                        seen_stocks:
                result.append({
                        'Date': date,
                        'Start Time': start_time,
                        'End Time': row['datetime'].time(),
                        'Stock Code': stock_code,
                        '25-Minute Return': value
                    })
            seen_dates.add(date)
            seen_stocks.add(stock_code)
            seen_start_times.add(start_time)
            if len(result) >= sample_nums:
                break
    if len(result) >= sample_nums:
        break

result_df = pd.DataFrame(result)
```

Figure 15: Generated code to filter out the "Sharp Drop" case

---

**Scenario: Sharp Drop**

---

**Data Description**: The input data is in CSV format with the following information.

- The first column "date" represents the trading date.

- The second column "minute" represents the time.

- Each subsequent column corresponds to an instrument, with the value in each cell representing the return of the instrument for the given minute compared to the previous minute.

**Output Description**: Please identify and provide 30 samples where a stock drops sharply within a 25-minute window. For each sample, include the following details:

1. Date.

2. Start and end minute of the 25-minute window.

3. Stock code.

4. The return of the 25-minute interval.

**Constraints on Output:**

1. Ensure that the 25-minute cases do not contain duplicate stock codes and datetimes. Each sample should be selected from different trading days.

2. Ensure that each 25-minute interval is within the same trading day.

3. You can use `groupby('datetime').rolling(25).sum()` to convert 1-minute-level returns to 25-minute-level returns.

4. The begin and end times of the 25-minute interval should be within trading hours, e.g., 9:30 AM - 11:30 AM and 1 PM - 3 PM.

---

Table 3: Prompt used for generating code in the "Sharp Drop" scenario

- **Sharp Drops**: Simulating sharp declines to understand market reactions to negative events, assess risk management strategies, and evaluate market liquidity.

- **Sharp Rises**: Simulating sharp increases to capture market behavior during positive events, allowing traders to test profit-taking strategies and analyze upward trends.

- **Trend Reversals**: Simulating trend reversals to identify signals for entry or exit points and understand market reactions to trend shifts.

Fig. 16 displays real stock trends over the first 15 minutes and the stock trends generated by MarS for the last 10 minutes of a 25-minute period for these scenarios. Each row represents a scenario with three cases. The x-axis denotes time, and the y-axis indicates price. The blue line shows the replay price trajectory, and the orange line depicts the simulated price trajectory with confidence intervals. The results demonstrate MarS's capability to effectively generate diverse market scenarios, providing valuable insights for market participants.

# F    CONFIGURATIONS OF INPUT OVER DIFFERENT APPLICATIONS

As we abstract the mechanism of MarS as a conditional generation process in Sec.2, we summarize their input conditions over different applications in Table 4, and provide more detailed clarification.

**DES_TEXT** is a key component in the Conditional Trading Order Generation task, acting as a control mechanism for the "Conditional" aspect. It is designed to describe different market states under which we aim to generate trading orders. Examples of such market states include "sharp price decline" or "high market volatility". By incorporating **DES_TEXT**, we enable the generation process to adapt to varying market conditions, making the generated trading orders contextually relevant. More details on **DES_TEXT** are provided in Sec E.

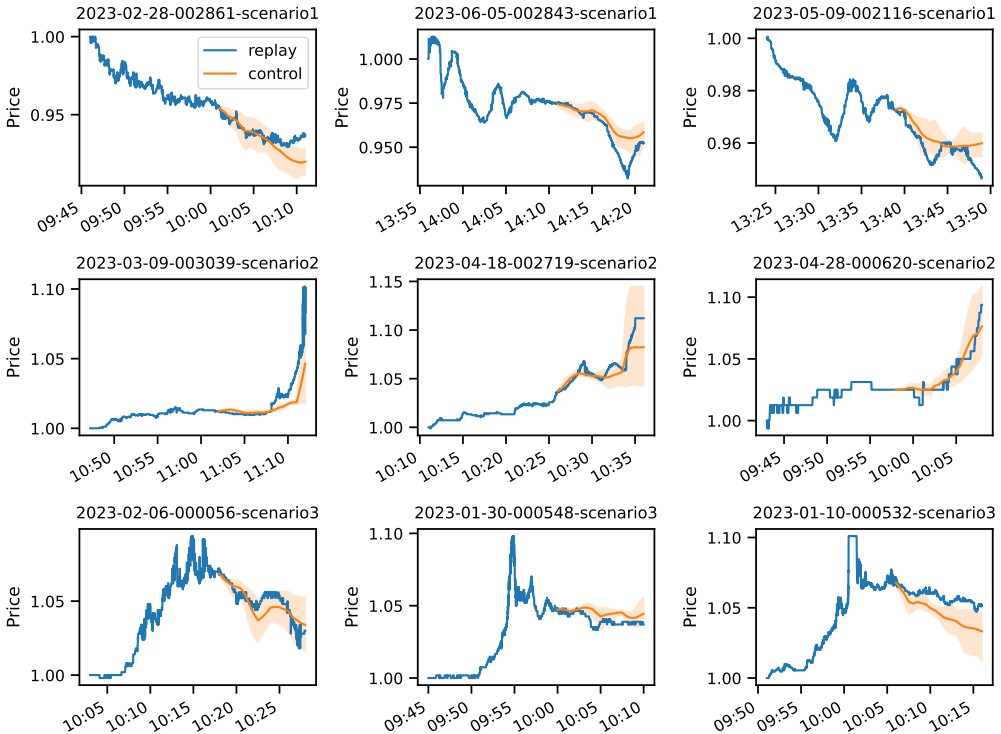

Figure 16: Case study for different scenario generation.

As for **MTCH_R**, it represents a comprehensive set of order-matching rules, for example, the widely used double auction mechanism. In real-world financial markets, the rules are specified and periodically adjusted by exchanges. In our simulation, these rules are governed by the Simulated Clearing House. We formulated **MTCH_R** as a hyperparameter to make the MarS framework adaptable to different markets and conditions. In the proposed paper, we set it as a series of standard settings of the default double auction. Expanding **MTCH_R** would reveal the full extent of an exchanges trading rules, encompassing many details that we have implemented in our code for the Simulated Clearing House.

Moreover, while the double auction mechanism is a common paradigm for the majority of global financial markets, there are variations in trading rules that differ across markets and periods. These include aspects such as price fluctuation limits, circuit breakers, and the distinction between call and continuous auction sessions. Our goal was to encapsulate these variations within the conditional trading order generation framework, ensuring the approach remains broadly applicable and flexible for different market scenarios.

| Applications | Input Conditions |
|---|---|
| Forecasting | $(x_0, \ldots, x_m), MTCH\_R$ |
| Detection | $(x_0, \ldots, x_m), MTCH\_R$ |
| "What if" Analysis | $[DES\_TEXT], (x_0, \ldots, x_m)^*, [(\dot{x}_{i+1}, \ldots, \dot{x}_{i+j})], MTCH\_R$ |
| RL Environment | $DES\_TEXT^*, (x_0, \ldots, x_m)^*, (\dot{x}_{i+1}, \ldots, \dot{x}_{i+j}), MTCH\_R$ |

Table 4: The summary of input conditions for order generation of different applications. $*$ means the condition is optional and $[\ ]$ indicates that either of the specified conditions should be chosen.

## G  DATA AND TECHNICAL DETAILS OF DETECTION

Traditional methods for detecting market abuse are time-consuming and challenging, and abnormal market states are often defined and detected based on the differences between current and historical

Table 5: Market manipulation samples collected from CSRC.

| Instrument | Start time | End time | Case Number |
|---|---|---|---|
| 300475 | 2017-03-07 | 2017-04-25 | [2020]No.92 |
| 002321 | 2017-04-17 | 2018-01-30 | [2024]No.44 |
| 300263 | 2017-05-17 | 2017-09-25 | [2023]No.36 |
| 300658 | 2019-02-13 | 2019-05-10 | [2023]No.25 |
| 300378 | 2019-03-14 | 2019-04-15 | [2021]No.116 |
| 300119 | 2019-04-01 | 2019-05-22 | [2021]No.116 |
| 002718 | 2020-06-04 | 2020-07-15 | [2022]No.64 |
| 300313 | 2020-08-19 | 2020-08-24 | [2021]No.76 |
| 002730 | 2020-12-15 | 2021-11-17 | [2024]No.23 |
| 002713 | 2022-05-05 | 2022-05-18 | [2024]No.47 |

market patterns. In this section, we take market manipulation as an example, and demonstrate how MarS could bring a new simulation-based paradigm to detection task.

Table 5 shows the market manipulation samples collected from China Securities Regulatory Commission (CSRC). The data encompass a total of 10 stocks, which have never been included in datasets used for our model training. For each stock, we gathered samples from an equal number of trading days before and after the manipulation occurred for comparison. There are 522 trading days for each period. For each trading day, we conducted simulations every 25 minutes and then calculated a series of stylized facts of the simulated and replay trajectories.

The spread is a key indicator of market liquidity, with a larger spread indicating poorer market liquidity. At time $t$, the spread $\delta$ is defined as: $\delta_t = a_t - b_t$, where $a_t$ is the best ask price and $b_t$ is the best bid price. The spread distribution is widely used in detection tasks in finance (Affleck-Graves et al., 2000; Vyetrenko et al., 2020).

As we evaluated MarS's realism in a normal market in Sec. 3, a straightforward principle for anomaly detection is that a quick drop in simulation realism metrics can serve as an initial indicator of potential anomalies. To verify it, we collected several market manipulation cases from CSRC[3]. For each stock, we collected replay samples before, during and after the manipulation, and conducted simulations by MarS simultaneously. Through calculating Distribution Similarity[4], we evaluate the similarity of spread distributions, which serves as a key indicator of market liquidity. This metric is used for comparison between replay and simulation.

Fig. 8 shows the varying spread distributions in different periods around manipulation. While MarS generally performs well to simulate the normal market, its performance drops during the manipulation, showing a heavier tail and a peak around $\delta = 1000$. These anomalies can be viewed as signals likely corresponding to market manipulation, where manipulators significantly impact liquidity, widening the spread. These anomalies, less frequent in normal markets, lead to a performance drop in MarS, suggesting a new detection approach by monitoring such similarity drops. Consequently, MarS can help investors avoid anomalies and assist financial institutions in maintaining market stability.

It is important to note that a single anomaly does not conclusively indicate market manipulation. Instead, it serves as an initial signal that requires further holistic assessment, combining multiple metrics to ensure robust conclusions. The example provided serves as a representative illustration of our approach. Our primary objective in this experiment was to demonstrate the paradigm shift MarS offers in market manipulation detection.

---

[3]http://www.csrc.gov.cn
[4]https://en.wikipedia.org/wiki/Overlap_coefficient

## H CONFIGURABLE TWAP STRATEGY AND TRADING AGENT

### H.1 INTRODUCTION OF TWAP STRATEGY

The Time-Weighted Average Price (TWAP) algorithm executes large trade volumes while minimizing market impact over a specified time frame. The TWAP strategy divides the total volume to be traded into equal parts that are executed at regular intervals. This strategy consists of two distinct phases within each interval: the passive period and the aggressive period. Key configurations include:

- **Maximum Passive Volume Ratio (PVR)**: During the passive period, the strategy places orders at the current bid price (bid1) with a volume determined by the PVR, aiming to fill orders without significantly altering the market price. A PVR of 0 indicates no passive volume during the passive period.
- **Aggressive Price (AP)**: If passive trading does not achieve the expected volume, the strategy enters an aggressive phase, placing additional orders at a more aggressive price (AP) to ensure the desired volume is executed. An AP of 0 means no aggressive order during the aggressive period.

By balancing passive and aggressive trading, the TWAP strategy aims to execute large orders efficiently while controlling market impact.

Taking the buying task as an example, our configurable TWAP strategy is shown as below:

---

**Algorithm 1** Configurable Time-Weighted Average Price (TWAP) Strategy for Buying.

---

**Input:** Total Volume $V$, Execution Time $T = 5$ minutes, Split Interval $\Delta t = 30$ seconds, Maximum Passive Volume Ratio $PVR$, Aggressive Price $AP$ (ask1, ask2, ..., ask5)
**Output:** Executed Orders
**Initialization:**
    1. Split the total volume $V$ into 10 equal parts. Each part $K = V/10$ is expected to be executed in $\Delta t = 30$ seconds.

**For each interval $i$ from 1 to 10:**

    1. **Passive Period:** (First 25 seconds of each interval)

        (a) Cancel all non-bid1 volumes.
        (b) Submit a passive order with max volume $PVR \times V$ and price bid1.
        (c) Wait for 25 seconds.

    2. **Aggressive Period:** (Last 5 seconds of each interval)

        (a) If the current executed volume lags behind the expected volume:
            • Calculate the extra volume $E$ to be executed.
            • If the available volume is insufficient, cancel existing passive orders as needed.
            • Submit an aggressive order with volume $E$ and price $AP$.
        (b) Wait for 5 seconds.

---

### H.2 TRAINING OF TWAP TRADING AGENT WITH RL

For the trading agent training with RL, we can adjust the maximum passive volume ratio (PVR) from $\{0, 0.1, \ldots, 1\}$ and aggressive price (AP) in $\{0, 1, 2, 3, 4, 5\}$ for TWAP Strategy. We used a batch size of 8192 and a learning rate of $4 \times 10^{-5}$. The trading model was updated using a simple policy gradient algorithm (Sutton & Barto, 2018). The performance metric is the price advantage over our best-configured TWAP agent (L1-P0.9), measured in basis points (BP, 1/10000).

## I EVALUATION OF CONT'S 11 STYLIZED FACTS

### I.1 SUMMARY

Stylized facts are high-level summaries of empirical characteristics in financial markets, essential for assessing the realism of market simulations. In this section, we evaluate the 11 stylized facts identified by Cont (2001) using historical and simulated order sequences.

To rigorously test these facts, we simulated 11,591 trajectories for the top 500 liquid stocks in the Chinese market, from March 9, 2023, to July 12, 2023. Table 6 compares the presence of these facts in both historical and simulated data. The **Historical** column indicates observation in real data, while the **Simulated** column assesses their presence in simulated data. Key findings include:

- Nine out of the 11 stylized facts are observed in both historical and simulated data. However, *Gain/loss asymmetry* and *Leverage effect* are not present, possibly reflecting modern market shifts. Studies such as Ratliff-Crain et al. (2023) note similar absences in the modern U.S. Dow 30 stocks.
- All 11 facts show similar patterns between simulated and historical sequences, showcasing the model's strong capability in generating realistic order sequences.

Note that merely evaluating stylized facts does not fully assess financial market simulation quality. Further evaluations for **in-context** generation, such as forecasting (Section 4.1) and quantitative analysis of stylized facts (Section J), are crucial.

| Fact # | Fact Name | Historical | Simulated |
|--------|-----------|:----------:|:---------:|
| 1 | Absence of autocorrelations | × | × |
| 2 | Heavy tails | × | × |
| 3 | Gain/loss asymmetry | | |
| 4 | Aggregational Gaussianity | × | × |
| 5 | Intermittency | × | × |
| 6 | Volatility clustering | × | × |
| 7 | Conditional heavy tails | × | × |
| 8 | Slow decay of autocorrelation in absolute returns | × | × |
| 9 | Leverage effect | | |
| 10 | Volume/volatility correlation | × | × |
| 11 | Asymmetry in timescales | × | × |

Table 6: Presence of Stylized Facts in Historical and Simulated Order Sequences. All facts are present in both historical and simulated data, except for *Gain/loss asymmetry* and *Leverage effect*.

## I.2 DEFINITIONS OF STYLIZED FACTS

The 11 stylized facts from Cont (2001) are:

1. **Absence of autocorrelations**: "(linear) autocorrelations of asset returns are often insignificant, except for very small intraday time scales (20 minutes) for which microstructure effects come into play."

2. **Heavy tails**: "the (unconditional) distribution of returns seems to display a power-law or Pareto-like tail, with a tail index which is finite, higher than two and less than five for most data sets studied. In particular this excludes stable laws with infinite variance and the normal distribution. However the precise form of the tails is difficult to determine."

3. **Gain/loss asymmetry**: "one observes large drawdowns in stock prices and stock index values but not equally large upward movements."

4. **Aggregational Gaussianity**: "as one increases the time scale $t$ over which returns are calculated, their distribution looks more and more like a normal distribution. In particular, the shape of the distribution is not the same at different time scales."

5. **Intermittency**: "returns display, at any time scale, a high degree of variability. This is quantified by the presence of irregular bursts in time series of a wide variety of volatility estimators."

6. **Volatility clustering**: "different measures of volatility display a positive autocorrelation over several days, which quantifies the fact that high-volatility events tend to cluster in time."

7. **Conditional heavy tails**: "even after correcting returns for volatility clustering (e.g. via GARCH-type models), the residual time series still exhibit heavy tails. However, the tails are less heavy than in the unconditional distribution of returns."

8. **Slow decay of autocorrelation in absolute returns**: "the autocorrelation function of absolute returns decays slowly as a function of the time lag, roughly as a power law with an exponent $\beta \in [0.2, 0.4]$. This is sometimes interpreted as a sign of long-range dependence."

9. **Leverage effect**: "most measures of volatility of an asset are negatively correlated with the returns of that asset."

10. **Volume/volatility correlation**: "trading volume is correlated with all measures of volatility."

11. **Asymmetry in time scales**: "coarse-grained measures of volatility predict fine-scale volatility better than the other way round."

### I.3 EVALUATION OF STYLIZED FACTS

This subsection summarizes the evaluation results for each stylized fact. Initially, each instrument is assessed individually, and the results are then aggregated across all instruments to obtain an average. A 95% confidence interval is shown for line plots, and quantiles are displayed for the box plot.

**Absence of autocorrelations**: We computed the autocorrelation of returns using both the last and mean trade prices per minute. Fig. 17a and 17b illustrate that autocorrelations decay quickly after one minute. Using the last trade price shows negative autocorrelation at lag 1 due to the "bid-ask bounce", as noted in Ratliff-Crain et al. (2023). Conversely, the mean trade price shows positive autocorrelation, indicating short-term momentum. For consistency with Ratliff-Crain et al. (2023), we use the last trade price for subsequent evaluations.

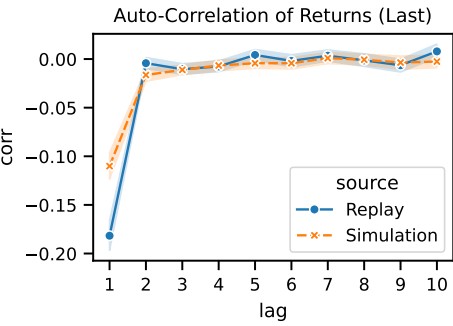 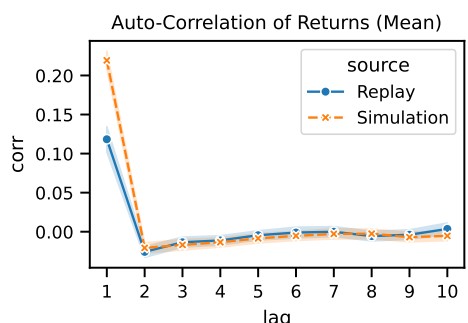

(a) Absence of autocorrelations (Last Price)      (b) Absence of autocorrelations (Mean Price)

Figure 17: Absence of autocorrelations. (**a**) Using *last* trade price. (**b**) Using *mean* trade price. Both show rapid decline after 1 minute.

**Heavy tails** and **Aggregational Gaussianity**: Kurtosis of returns for various intervals was calculated. Positive kurtosis indicates sharper peaks and heavier tails than normal distribution. Fig. 18a shows that return distributions exhibit heavy tails. Distributions trend towards normality as intervals extend from 1 to 20 minutes, aligning with Aggregational Gaussianity.

**Conditional heavy tails**: Volatility varies throughout the trading day, peaking at open and close. After normalizing returns by minute-specific volatility and computing kurtosis, Fig. 18b shows that normalized returns still exhibit heavy tails, though less pronounced than unconditional returns in Fig. 18a, consistent with Conditional heavy tails.

**Gain/loss asymmetry**: Positive skewness of returns (Fig. 19a) suggests a deviation from Cont's original description.

**Volatility clustering** and **Slow decay of autocorrelation in absolute returns**: Autocorrelation of absolute returns for different intervals shows slow decay in Fig. 19b. Considering absolute returns as volatility Mller et al. (1997), this also illustrates volatility clustering.

**Intermittency**: Following Ratliff-Crain et al. (2023), extreme returns are defined as the 99% quantile of absolute returns. The Fano factor, used to verify Poisson distribution adherence, exceeded

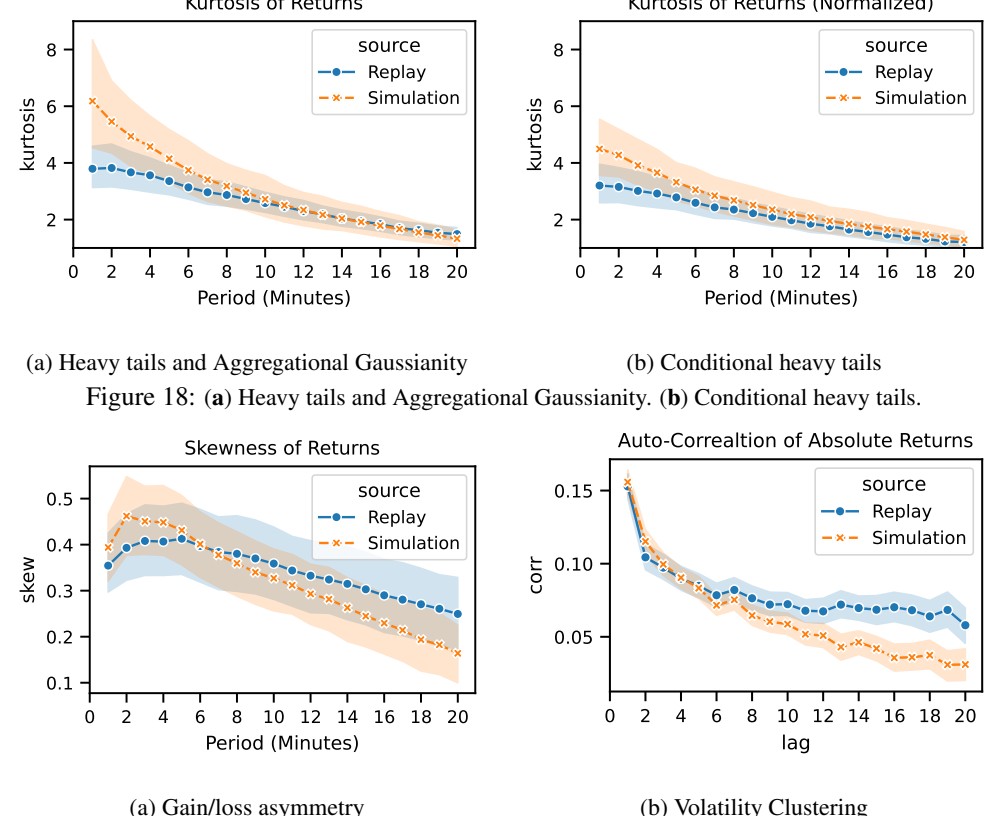

(a) Heavy tails and Aggregational Gaussianity    (b) Conditional heavy tails

Figure 18: (**a**) Heavy tails and Aggregational Gaussianity. (**b**) Conditional heavy tails.

(a) Gain/loss asymmetry    (b) Volatility Clustering

Figure 19: (**a**) Gain/loss asymmetry: right-skewed distribution. (**b**) Volatility Clustering: slow decay of absolute return autocorrelation.

1, indicating higher variability (Fig. 20a). This, along with heavy tails and volatility clustering, confirms Intermittency.

**Leverage effect**: Return and lagged volatility correlation is slightly positive (Fig. 20b), contrary to Cont's description.

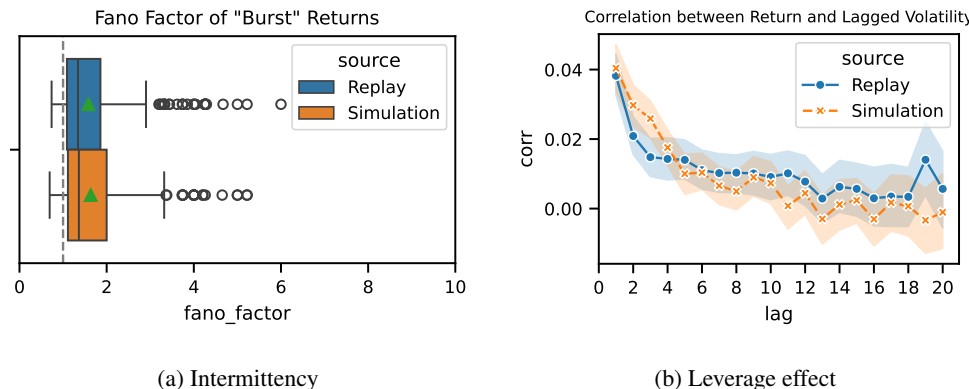

(a) Intermittency    (b) Leverage effect

Figure 20: (**a**) Intermittency: Fano factor exceeds 1, indicating high variability. (**b**) Leverage effect: slightly positive correlation between return and lagged volatility.

**Volume/volatility correlation**: Positive correlation between volume and lagged volatility is evident (Fig. 21a).

**Asymmetry in timescales**: Following Takahashi et al. (2019a), we assessed correlation between fine- and coarse-grained volatility across lags from -10 to 10 minutes. Fig. 21b shows significant negative asymmetry, consistent with Takahashi et al. (2019a) and Mller et al. (1997).

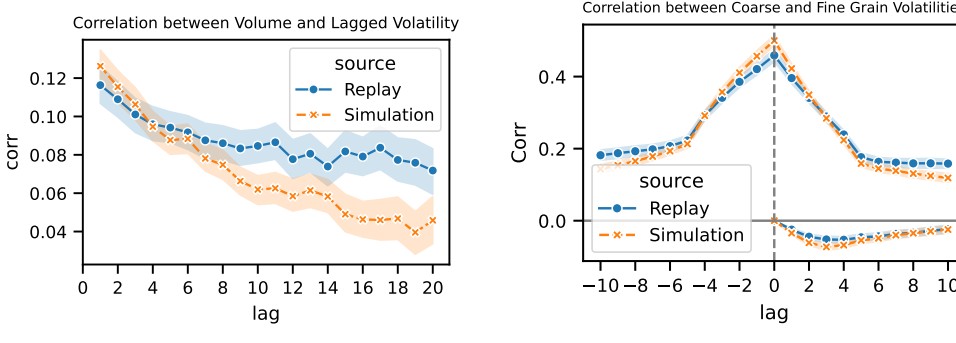

(a) Volume/volatility correlation        (b) Asymmetry in timescales

Figure 21: (**a**) Volume/volatility correlation: positive correlation. (**b**) Asymmetry in timescales: significant negative asymmetry observed.

## J  QUANTITATIVE ANALYSIS OF STYLIZED FACTS

To ensure experiments are comparable across runs, we quantify the stylized facts with two metrics:

- **Distribution Similarity**: We calculate the overlap coefficient between the empirical distribution of the stylized fact and the simulated distribution. A higher score indicates a higher similarity in the overall distribution.

- **Accuracy (3-Class)**: We classify one stylized fact value into three classes based on replay data: low, medium, and high, ensuring similar probabilities for each class over the simulation period. We then compare the stylized fact value between simulation and replay and calculate the accuracy of the classification. This metric measures our capability for in-context prediction.

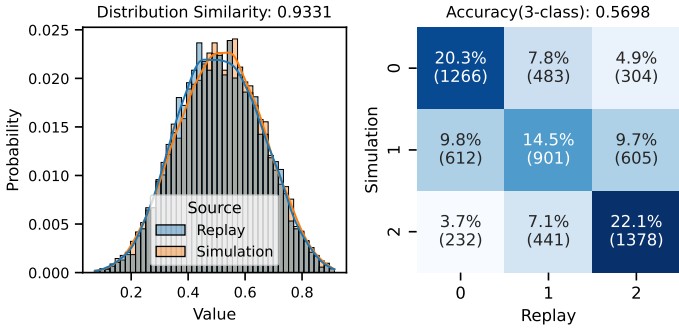

Figure 22: Stylized Fact Analysis: Buy Order Ratio. This metric assesses the proportion of buy to buy+sell orders, capturing market dynamics that may influence the market trend.

We show an example for the Buy Order Ratio in Fig. 22: we calculate the buy order ratio for each minute and then compare the distribution of the ratio between simulation and replay data. In summary, we achieve a high score for the overall distribution similarity and an acceptable 3-class classification considering the nuances of market dynamics. We list the full quantitative results in Table 7.

## K  MARKET IMPACT

We give a detailed introduction and discussion on interactive simulation and market impact analysis.

**Market Impact Generation:** We generate market impact data using the TWAP strategy with four different configurations: L1-P0.1, L1-P0.9, L5-P0.1, and L5-P0.9. The configuration name LX-PY indicates that the aggressive price (AP) is askX and the maximum passive volume ratio (PVR) is

| Name | Distribution Similarity | Accuracy (3-Class) |
|---|---|---|
| Volatility | 0.872 | 0.516 |
| Spread | 0.970 | 0.729 |
| Mean Order Volume | 0.957 | 0.776 |
| Aggressive Order Ratio | 0.920 | 0.525 |
| Buy Order Ratio | 0.933 | 0.570 |
| 1-Min Return | 0.956 | 0.684 |
| 2-Min Return | 0.936 | 0.625 |
| 3-Min Return | 0.924 | 0.583 |
| 4-Min Return | 0.914 | 0.548 |
| 5-Min Return | 0.908 | 0.531 |

Table 7: Summary of stylized facts. The prediction for 1 to 5-Min Return is aggregated from 128 rollouts for each initial time point.

Y. These agents are assigned to buy varying volumes over 5 minutes with different instructions and starting times. We explored the market impact generated by these trading agents from 624k simulated trading trajectories.

**Further analysis of synthetic market impact:** Beyond the verification of the Square-Root-Law, we apply further analysis on synthetic market impact data. The key findings are summarized as follows:

- Agents with more aggressive configurations (L5-P0.1 and L5-P0.9) are expected to exhibit a larger market impact and achieve a higher fulfillment rate. Our simulations quantify their differences and confirm these assumptions, as illustrated in Fig. 23a.

- The agents generate both short-term and long-term market impacts in MarS, as shown in Fig. 23b, similar to observations studied in previous empirical work (Bacry et al., 2014; Donier et al., 2015b). We also observe that agents with a larger passive volume ratio generate less momentum after trading ends.

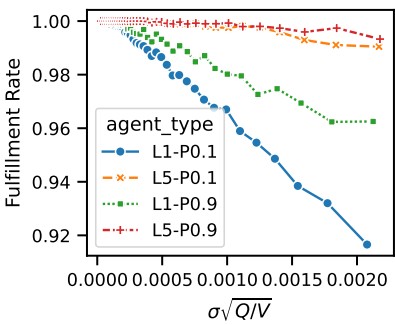
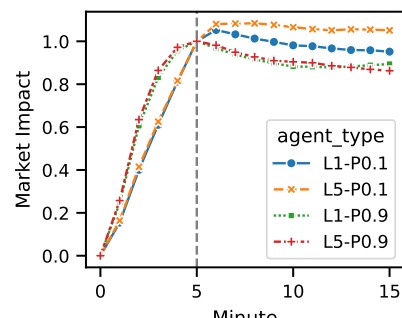

(a) Fulfillment rate of different agents  (b) Short-term and long-term market impact

Figure 23: Further investigation of synthetic market impact

These findings confirm the reliability and convenience of using synthetic data from MarS, allowing for in-depth exploration of market dynamics without the cost, risk, and time constraints associated with real-world experiments.

**New factors of Market Impact:** The new three factors $\{resiliency, LOB\_pressure, LOB\_depth\}$ are defined as below:

$$resiliency = 1 - \log(|pre\_trading\_moment|) \tag{5}$$

$$LOB\_pressure = (\alpha * agent\_trans\_ask + (1 - \alpha) * agent\_trans\_bid) * LOB\_imb_{last\text{-}pre\text{-}min} \tag{6}$$

$$LOB\_depth = \log(\beta * LOB\_ask\_volume_{last\text{-}pre\text{-}min} + (1 - \beta) * LOB\_bid\_volume_{last\text{-}pre\text{-}min}), \tag{7}$$

where:

$$pre\_trading\_moment = \frac{\sum_{t_0}^{last\text{-}pre\text{-}min-1} \gamma_t * mid\_price_t}{mid\_price_{last\text{-}pre\text{-}min}} - 1 \tag{8}$$

$$agent\_trans\_ask = \frac{\sum_{t=trade\_start}^{trade\_end} agent\_trans\_volume_t}{\sum_{t=trade\_start}^{trade\_end} agent\_trans\_volume_t + LOB\_ask\_volume_{last\text{-}pre\text{-}min}} \tag{9}$$

$$agent\_trans\_bid = \frac{\sum_{trade\_start}^{trade\_end} agent\_trans\_volume_t}{\sum_{trade\_start}^{trade\_end} agent\_trans\_volume_t + LOB\_bid\_volume_{last\text{-}pre\text{-}min}} \tag{10}$$

$$LOB\_imb_{last\text{-}pre\text{-}min} = \frac{|LOB\_ask\_volume_{last\text{-}pre\text{-}min} - LOB\_bid\_volume_{last\text{-}pre\text{-}min}|}{LOB\_ask\_volume_{last\text{-}pre\text{-}min} + LOB\_bid\_volume_{last\text{-}pre\text{-}min}}, \tag{11}$$

and $\alpha, \beta, \{\gamma_t\}$ are the hyper-parameters with constrain: $\alpha \in (0,1)$, $\beta \in (0,1)$, $\gamma_t \in (0,1)$ for any $t$, and $\sum_{t_0}^{last\text{-}pre\text{-}min-1} \gamma_t = 1$. *last-pre-min* means the last minute before the agent starts to trade. *LOB_ask_volume* and *LOB_bid_volume* are the ask and bid volumes of LOB. $agent\_trans\_volume_t$ is the transaction volume of the agent at time $t$. $mid\_price_t$ is the mid-price at time $t$.

The relationship between market impact and factors *LOB_pressure*, and *LOB_depth* is shown in Fig. 24.

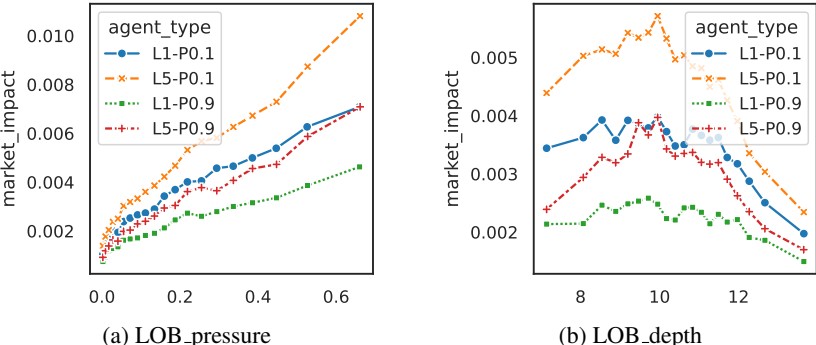

(a) LOB_pressure           (b) LOB_depth

Figure 24: Effects of new factors on market impact.

We also investigate the correlation between three new factors and the Square-Root-Law factors: $sqrt(Q/V)$ and volatility $\sigma$ in Fig. 25. It is clear that the correlation scores of those factors are relatively low.

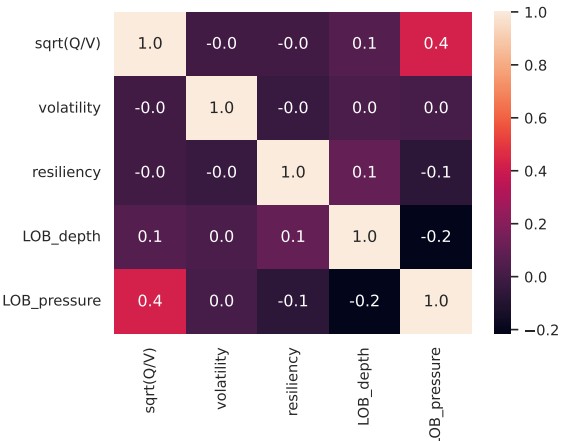

Figure 25: Correlation matrix of Square-Root-Law factors and three new factors.

**Dynamics of long-term Market Impact:** For equation 14 used to model the long-term market impact, we set two decay functions: $F^{decay}(t) = [\frac{1}{t}, \frac{1}{\sqrt{t}}]$ and seven factors: $\{\sqrt{\frac{Q}{V}}, mid\text{-}price, agent\_replay, agent\_rollout, LOB\_depth, LOB\_pressure, resiliency\}$. *mid-price* is

the mid-price before trading. *agent_rollout* and *agent_replay* are defined as below:

$$agent\_rollout = \frac{\sum_{trade\_start}^{trade\_end} agent\_trans\_volume_t}{total\_transaction\_volume\_of\_rollout} \quad (12)$$

$$agent\_replay = \frac{\sum_{trade\_start}^{trade\_end} agent\_trans\_volume_t}{total\_transaction\_volume\_of\_replay} \quad (13)$$

The training process is based on the synthetic long-term market impact generated by the TWAP agent ($L1 - P0.1$). We use torch-diff Chen (2018) to optimize $W$, where the objective is set as the MSE reconstruction loss along with the L1 regularization.

After training, we illustrate the auto-correlation of the synthetic market impact decay, the trajectories predicted by the learned ODE, and the base ODE from empirical formulas (Gatheral et al., 2011; Curato et al., 2017) in Fig. 26.

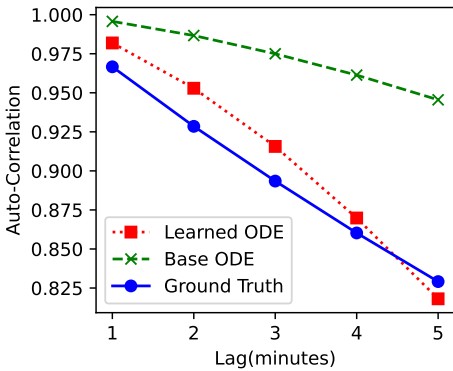

Figure 26: Auto-correlation of long-term market impact with learned ODE and base-ODE.

For the base-ODE used as a baseline in Fig. 26, we use the basic form of the Square-Root Process (Gatheral, 2010), which is defined as:

$$\frac{dY(t)}{dt} = \sigma \sqrt{\frac{Q}{V}} \frac{1}{\sqrt{t}} \quad (14)$$

where $\sigma$ is the volatility, $Q$ is the trading volume, and $V$ is the total market volume.

## L   COMPARISON OF DEEPLOB AND MARS/LMM IN FORECASTING TASKS

| Aspect | DeepLOB | MarS/LMM |
|---|---|---|
| Applicable Tasks | Task specific forecasting. | General forecasting through simulation. |
| Input Features | Limit order book (LOB) data. | High-frequency order-level data. |
| Model | Small, handcrafted, and not scalable | Large-scale foundation model. |
| Prediction | Single-step or fixed-length. | Multi-step, sequence generation. |

Table 8: Comparison of DeepLOB and MarS/LMM in forecasting tasks.

Table 8 compares DeepLOB and MarS/LMM in forecasting tasks, emphasizing their distinct approaches and capabilities. DeepLOB is designed for specific forecasting tasks, trained on fixed step forecasting, and uses Limit Order Book (LOB) data as input. It features a relatively small, handcrafted model for LOB forecasting, which is hard to scale up, and provides single-step predictions for fixed-length forecasting, such as price changes after 100 orders or 1 minute. In contrast, MarS is designed for market simulation, capable of performing general forecasting through simulation, and uses fine-grained order sequence data as input. It is powered by large foundation models trained on large-scale order sequence data and offers simulation with multi-step generation.

