# OpenReview forum: "MarS: a Financial Market Simulation Engine Powered by Generative Foundation Model"
_ICLR.cc/2025/Conference — ICLR 2025 Poster_

### Official Review · Reviewer_dx5Y · 2024-10-30

**Soundness:** 2
**Presentation:** 4
**Contribution:** 3
**Rating:** 8
**Confidence:** 4

**Summary:**

This paper presents a market simulation environment driven by a generative model that dynamically produces new market orders, based on the current market context and the historical order flow. This enables a responsive, interactive simulation of market dynamics. The model aims at supporting various applications, including forecasting, anomaly detection, market analysis, and agent training.

The core generative model, named the Large Market Model (LMM), is designed on the idea of a "foundation model for financial market simulation, akin to language modeling in the digital world".

**Strengths:**

- The paper addresses an important and complex challenge, moving a step towards a more general and powerful generative model for limit order book simulation.
- The authors use for the first (to the best of my knowledge) the LLaMA2 architecture for order flow generation, showing also the potential of this architecture with more data and model parameters (Fig.2)
- The downstream tasks are quite interesting and useful.
- The paper is well written and easy to follow.

**Weaknesses:**

Overall, I believe the paper has significant merit and I tend to recommend acceptance (and increase my score). However, I have some concerns, particularly regarding reproducibility. I would encourage the authors to make efforts to improve this aspect of their framework.

Below, I outline my main concerns and questions:
- Key details needed for reproducibility, such as code or datasets, are missing.
- Given that this is a conference on representation learning, more discussion on the embedding of orders and the LOB state would be valuable. For instance, are all possible order prices linearly discretized into the range [0, 32)? Are price outliers excluded? does the model use absolute prices or the distance from the midprice?
- What are the training and simulation times? The latter is particularly crucial for the proposed tasks, as generating a few hours of price paths requires thousands of generated orders.
- A discussion of limitations is missing. For example, unlike text with a relatively fixed vocabulary, a fixed embedding for orders might reduce adaptability across different markets?
(see also questions)

Minor:
- Some essential details should appear in the main paper, such as the dataset used and the authors’ use of a LLAMA2 architecture.
- In line 40, when the authors briefly discuss existing market simulation models, it would be helpful to include citations. This will guide readers interested in statistical or agent-based models to relevant literature.
- There is occasional overuse of non-technical terms, such as “gold mine of data,” and some paragraphs (e.g., lines 236 and 240), which are also capitalized. Are these intended as citations?
- In paragraph 2.1.2, clearer spacing or breaks between components would improve readability.
- Typo on line 384 (“classesup…”).
- The diffusion model in Figure 1 is not explained or discussed.

**Questions:**

- What does "MTCH R" refer to exactly? It’s not clear how a conditional model should interpret a constraint on the "feasible space for each order".

- Is the "SIMULATED CLEARING HOUSE" simply an exchange engine simulating a double auction market, or is it something else? It seems to be the latter.

- For the order-batch model, which generates orders at the end of each minute, is this the smallest reaction time of the simulation in response to orders? Or does the model plan a minute ahead (a batch of orders) and then refine (by conditioning) that batch using the Order model? How exactly does the condition in Figure 3 function here?

- In Figure 5a, the market replay appears to be shifted (due to buy orders). Is this intentional? Using the replay series as a control? I suggest showing this experiment with a shaded area over multiple simulations with similar buy and sell orders (similar to Figure 17).

- Why is MarS assumed to have a "real" transition distribution P(s | s, a) in Table 1?

- In Section 4.2, the detection approach is interesting, but I’m curious whether the observed difference in distribution similarity (computed how?) is statistically significant. Are stressful trading days or shocks interpreted as manipulation as well? Before claiming this approach is feasible, it would be helpful to see the average difference in spread distribution over multiple days or periods.

- In Figure 19b, the market impact appears to be permanent. Is it true? What happens if we extend the simulation period?

- How does the model adapt to different markets (without re-training)? For example, Chinese markets generally differ significantly from U.S. markets in terms of liquidity and frequency.

---

> ### Author Response · Authors · 2024-11-21
>
> We thank the reviewer for the careful review. We address the comments below (C: comments; R: responses):
>
> **C1: Given that this is a conference on representation learning, more discussion on the embedding of orders and the LOB state would be valuable. For instance, are all possible order prices linearly discretized into the range [0, 32)? Are price outliers excluded? Does the model use absolute prices or the distance from the midprice?**
>
> **R1:**
>
> Thank you for your valuable suggestion. We agree that more discussion on the embedding of orders and the LOB state would enrich the paper, and we plan to include additional details in the updated version.
>
> Regarding your specific question on discretization, we appreciate your attention to this detail. For order price (the price distance relative to the LOB mid-price), volume, and interval, we build a converter to discretize these values based on real data distribution. More specifically, during the converter building period, we sort the data and split it into `num_bins` slots, recording the boundary and samples of each slot.
>
> For instance, order prices are discretized into 32 slots: 30 slots represent the differences from the mid-price, distributed across the range. The discretization is non-linear—slots closer to the mid-price are more granular, allowing for finer resolution in areas where the majority of trades typically occur. Two additional slots are for outliers on either end. For the corner case where the price lies outside the defined range, it is assigned to one of the outlier slots. This ensures that all possible values are effectively captured. To provide further clarity, we have included the distribution of these 32 slots below:
>
>
>
> | Bin Index | Probability | Number of Price Units from Mid |
> | --------- | ----------- | ------------------------------ |
> | 0         | 0.013077    | (-∞, -57]                      |
> | 1         | 0.012817    | (-57, -32]                     |
> | 2         | 0.013559    | (-32, -21]                     |
> | 3         | 0.012746    | (-21, -15]                     |
> | 4         | 0.014415    | (-15, -11]                     |
> | 5         | 0.010942    | (-11, -9]                      |
> | 6         | 0.015833    | (-9, -7]                       |
> | 7         | 0.011042    | (-7, -6]                       |
> | 8         | 0.015044    | (-6, -5]                       |
> | 9         | 0.020247    | (-5, -4]                       |
> | 10        | 0.029436    | (-4, -3]                       |
> | 11        | 0.044273    | (-3, -2]                       |
> | 12        | 0.086925    | (-2, -1]                       |
> | 13        | 0.217492    | (-1, 0]                        |
> | 14        | 0.216176    | (0, 1]                         |
> | 15        | 0.068056    | (1, 2]                         |
> | 16        | 0.03482     | (2, 3]                         |
> | 17        | 0.022385    | (3, 4]                         |
> | 18        | 0.016058    | (4, 5]                         |
> | 19        | 0.013052    | (5, 6]                         |
> | 20        | 0.009246    | (6, 7]                         |
> | 21        | 0.006952    | (7, 8]                         |
> | 22        | 0.005971    | (8, 9]                         |
> | 23        | 0.009713    | (9, 11]                        |
> | 24        | 0.010313    | (11, 14]                       |
> | 25        | 0.009765    | (14, 18]                       |
> | 26        | 0.00931     | (18, 23]                       |
> | 27        | 0.010836    | (23, 31]                       |
> | 28        | 0.009321    | (31, 41]                       |
> | 29        | 0.010288    | (41, 58]                       |
> | 30        | 0.00973     | (58, 92]                       |
> | 31        | 0.01016     | (92, ∞]                        |
>
> In this table:
> - **Probability** refers to the proportion of data that falls within each bin relative to the entire dataset.
> - **Number of Price Units from Mid** refers to the distance from the mid-price measured in the smallest price increment.
>
> Additionally, we define the mid-price as:
> `mid-price = bid1 + (ask1 − bid1) // 2`
>
> We hope this explanation clarifies our embedding process. Further details will be included in the revised version to address these aspects comprehensively.

---

> ### Author Response · Authors · 2024-11-21
>
> **C2: What are the training and simulation times?**
>
> **R2:**
>
> Thank you for your question.
>
> To provide a detailed example, training the order model, which consists of 1.02 billion parameters, takes approximately 3 days using 4 Nvidia H100 GPUs and DeepSpeed stage 2 optimization. This training process involves 16 billion tokens.
>
> For simulation, the number of orders generated varies across different stocks, with an average of 90 orders per minute in our dataset. In our current experimental setup, simulating 10 minutes with an average of 900 orders takes about 1.1 minutes using one Nvidia A100 GPU. For batch experiments, we utilize 128 Nvidia A100 GPUs with batch inference to accelerate the simulation process for different instruments and random seeds.
>
> In the future, we aim to significantly boost simulation performance by leveraging quantization[1] and advanced serving systems such as Triton[2] and vLLM[3], which are not included in our current experimental setup.
>
> Reference:
>
> [1] https://huggingface.co/docs/optimum/en/concept_guides/quantization
>
> [2] https://github.com/triton-lang/triton
>
> [3] https://github.com/vllm-project/vllm
>
> **C3: Discussion on limitation**
> **R3:**
>
> Thank you for your comment regarding the limitations of MarS. We appreciate the opportunity to reflect on the current constraints and outline future directions for improvement.
>
> 1. **Multi-Modality Limitations**:
>    Currently, MarS relies heavily on historical orders and structured market data as inputs and control signals. While we support natural language inputs through the `DES_TEXT` feature, these inputs are first translated into order sequence patterns before being processed by the system. This approach limits the ability of MarS to fully integrate and model multi-modal market information, such as news sentiment, macroeconomic indicators, or other unstructured data sources. Enhancing the system's multi-modality capabilities is a key area of future work.
>
> 2. **Modeling Across Assets**:
>    At this stage, MarS is designed to model single assets and their corresponding Limit Order Books (LOB) during inference. This single-asset focus constrains the system's ability to capture cross-asset interactions, such as arbitrage opportunities or correlated trading behaviors. Addressing this limitation by extending MarS to simultaneously model multiple assets and their interactions is an important goal for future iterations.
>
> 3. **Evaluation Time Horizon**:
>    Due to computational resource constraints and scalability considerations, our current evaluations are limited to a 30-minute window. While this timeframe allows us to effectively study short-term market dynamics, it restricts our ability to fully analyze long-term behaviors, such as extended periods of market impact or complex trading strategies over multiple hours. We aim to expand the evaluation period in future experiments to provide a more comprehensive assessment of MarS's capabilities.
>
> We are grateful for your feedback, as it highlights areas for improvement that align with our ongoing efforts to enhance MarS. We look forward to addressing these limitations in our future work.

---

> ### Author Response · Authors · 2024-11-21
>
> **C4: What does "MTCH R" refer to exactly? It’s not clear how a conditional model should interpret a constraint on the "feasible space for each order".**
> **R4:** Thank you for your question, please refer to our response R1 for reviewer nX8c.
>
> **C5: Clarify on "SIMULATED CLEARING HOUSE"**
> **R5:** Thank you for your suggestion, please refer to our response R1 for reviewer nX8c.
>
> **C6: For the order-batch model, which generates orders at the end of each minute, is this the smallest reaction time of the simulation in response to orders? Or does the model plan a minute ahead (a batch of orders) and then refine (by conditioning) that batch using the Order model? How exactly does the condition in Figure 3 function here?**
> **R6:**
>
> Great question! The working mechanism of the order-batch model aligns more closely with the second interpretation you suggested, where the intra-minute order-batch, as a target distribution, refines the order model, and the inter-minute order-batch is influenced by the orders completed in the previous minute (including model-generated orders and user-injected orders). Additionally, we would like to emphasize that the minute-level example is only illustrative—our framework is flexible and can adapt to other time periods as well.
>
> Our simulation framework operates at the finest granularity of ticks, meaning that the model can generate immediate responses to recent and user-submitted interactive orders. Within the current framework, the order-batch model generates a target distribution, which is then used to refine the logits produced by the order model. This refinement process is handled by a cross-attention ensemble model.
>
> Specifically:
> - The order model produces a logits vector representing the expected distribution of the next order.
> - To balance market impact and ensure controlled order generation, the ensemble model refines this distribution using the target order-batch distribution for minute t+1.
> - By sampling from the refined logits, an order is generated for minute t+1, which becomes the latest market order.
> - Throughout minute t+1, the order model and ensemble model iteratively generate orders to ensure that the simulation remains responsive and consistent with the defined distribution.
>
> This approach ensures both responsiveness at the tick level and the ability to incorporate higher-level distributional control from the order-batch model.
>
> **C7: Why is MarS assumed to have a "real" transition distribution \( P(s | s, a) \) in Table 1?**
> **R7:**
>
> Thank you for your insightful question about the "real" transition distribution \( P(s_{t+1} | s_t, a_t) \) in MarS, as highlighted in Table 1. We are happy to clarify this point.
>
> 1. **Limitations of Traditional Financial RL**:
>    In most existing financial reinforcement learning (RL) works, the environment is based on historical data, serving as an offline training setting. In such setups, the agent's actions cannot truly interact with the environment since historical data is fixed and immutable. As a result, the state transitions in these RL environments are effectively simplified to \( P(s_{t+1} | s_t) \), removing the explicit dependence on the agent's actions \( a_t \). This limitation significantly reduces the realism and effectiveness of these RL environments in modeling the actual impact of agent actions on the market.
>
> 2. **Interactive Design in MarS**:
>    In contrast, MarS enables agents to inject orders dynamically into the simulated market environment. These user-injected orders interact with the simulated background orders and directly influence the market state, including key features such as price trajectories and order book dynamics. This interactive capability ensures that the transition distribution \( P(s_{t+1} | s_t, a_t) \) reflects the realistic effects of agent actions on the environment. This interaction fundamentally differentiates MarS from traditional RL setups and provides a more faithful representation of real-world trading scenarios.
>
> Thus, we do not "assume" that MarS has a real \( P(s_{t+1} | s_t, a_t) \); rather, the interactive design of MarS inherently establishes this property. By allowing agent actions to dynamically influence the environment, MarS ensures that the transition distribution reflects a more realistic simulation of market dynamics.

---

> ### Author Response · Authors · 2024-11-21
>
> **C8: Whether the observed difference in distribution similarity (computed how?) is statistically significant. Are stressful trading days or shocks interpreted as manipulation as well?**
> **R8:** Thank you for your question, please refer to our response R5 for reviewer nX8c.
>
> **C9: In Figure 19b, the market impact appears to be permanent. Is it true? What happens if we extend the simulation period?**
> **R9:**
>
> Thank you for your insightful observation! We are glad to clarify this aspect.
>
> First, market impact is not permanent for most cases, and its resiliency depends heavily on the agent's settings, like target trading volume. Over the long term, after the trading period ends, it typically exhibits a slow decay pattern, eventually stabilizing as a near-flat curve. In Figures 5(a) and 19, the simulated period is relatively short, which limits the visibility of this decay pattern. However, as you observed, there is already evidence of a slight downward trend in the post-trading impact.
>
> A more detailed exploration of this phenomenon is presented in Section 4.3, where we study the Dynamics of Long-Term Market Impact. The learned ODE interaction weights, as shown in Figure 9(b), predominantly exhibit negative values, indicating a decay pattern over time. This supports the notion that market impact gradually diminishes rather than remaining constant.
>
> We acknowledge the value of extending the simulation period to further validate the long-term decay behavior. We plan to conduct additional simulations over longer time horizons to provide a more comprehensive understanding of market impact dynamics.
>
> **C10: How does the model adapt to different markets (without re-training)? For example, Chinese markets generally differ significantly from U.S. markets in terms of liquidity and frequency.**
> **R10:**
>
> Thank you for raising an excellent question! First, we regret to acknowledge that, in its current form, MarS cannot fully adapt to significantly different markets, without re-training. While the proposed MarS framework is designed to be general and flexible—thanks to its customizable matching rules and tokenization approach that can accommodate various market styles and trading rules—the model parameters and order embeddings remain tightly coupled to the characteristics of the specific market data used during training, limiting the model’s ability to generalize across markets without fine-tuning or re-training.
>
> In our future work, we plan to explore multi-market or cross-market pretraining strategies. This would involve leveraging diverse datasets from multiple markets during the training phase, enabling MarS to learn a broader set of patterns and features that could enhance its adaptability across different market environments.
>
> **C11: The diffusion model in Figure 1 is not explained or discussed.**
> **R11:** Thank you for your question, please refer to the response R4 for reviewer dWUs.
>
> **C12: Key details needed for reproducibility, such as code or datasets, are missing.**
> **R12:** Thank you for your question, please refer to the response R6 for reviewer CjFd.

---

> ### Author Response · Authors · 2024-11-25
> **Feedback on rebuttal**
>
> Dear Reviewer dx5Y,
>
> We greatly appreciate the time you took to review our paper. Due to the short duration of the author-reviewer discussion phase, we would appreciate your feedback on whether your main concerns have been adequately addressed. We are ready and willing to provide further explanations and clarifications if necessary.
>
> Thank you very much!

---

> ### Comment · Reviewer_dx5Y · 2024-11-25
>
> I thank the authors for taking the time to answer my concerns, and improve the paper presentation.
>
> The authors have not sufficiently justified the lack of reproducibility in this work. However, I still believe the paper has significant merits, and its acceptance at this conference would benefit the community. I still strongly encourage the authors to make every effort to improve the reproducibility sharing the code or pseudocode.
>
> Some points:
>
> -> In Table 1, I understand the authors' point, but I still think real P(.) is incorrect. I would rather explain better what you discussed here, or just say "better approximate/mimic real P(.)"
>
> -> For readability purpose: I'm not sure about introducing the terms "MTCH R" and "CLEARING HOUSE" in the main body of the paper, if not discussed enough or used in the experiments. I think it is more confusing than beneficial. I would rather suggest to add a sentence saying something like " [...] the matching engine is flexible to any rules [...] see Appendix X".
>
> -> The high simulation time is also related to the sampling procedure of the diffusion model? If so, I suggest the authors to check for denoising diffusion implicit models (DDIMs) or fast sampling procedures, along the quantization.

---

> > ### Author Response · Authors · 2024-11-26
> >
> > Dear Reviewer dx5Y,
> >
> > Thank you for your thoughtful comments and for acknowledging the improvements made to the paper presentation. We appreciate your constructive feedback and suggestions.
> >
> > Regarding the reproducibility of our work, we understand the importance of this aspect and are committed to enhancing it. We have already open-sourced some core components of MarS, including the market simulation engine along with several examples. We are actively working on releasing additional code and models, ensuring they meet the necessary standards for security, usability, and documentation. We will provide an update on the code's availability in the final version of the paper.
> >
> > For the usage of "P(.)" and the terms "MTCH R" and "CLEARING HOUSE", we agree that "better approximate/mimic real P(.)" is a more precise presentation, and adding an extra link to the appendix will make it clearer. We thank you for the valuable suggestions and will incorporate them in the latest version.
> >
> > Lastly, concerning the high simulation time, it is currently relatively slow mainly due to the low performance of the inference and serving system. We appreciate your suggestion to explore denoising diffusion implicit models (DDIMs) or fast sampling procedures, and we will investigate these methods to potentially improve the efficiency of our simulations.
> >
> > Thank you once again for your valuable feedback and support.

---

### Official Review · Reviewer_dWUs · 2024-11-03

**Soundness:** 3
**Presentation:** 4
**Contribution:** 3
**Rating:** 8
**Confidence:** 3

**Summary:**

This paper introduces a generative model designed to simulate realistic order-levels scenarios in financial markets. It uses a large language model to analyze the current market environment, taking this information within a proposed large market model. The model comprises two main components: an order-batch model that captures temporal patterns and an order model that addresses the sequential nature of market orders. These orders are then interactive with user's injection in a simulated in a simulated clearinghouse environment.

To validate the model, the paper conducts tests on stylized facts to ensure realistic simulation outcomes and applies the square-root law to assess interactive simulation capabilities. It also evaluates the model’s scalability in large-scale market scenarios.

Following validation, the model is applied in four downstream applications: forecasting, anomaly detection, what-if scenario analysis, and training datasets for agent-based models.

The paper contributes a model for simulating realistic financial markets, which has potential for application across various downstream scenarios.

**Strengths:**

The paper introduces a Large Market Mode and the MarS engine, which use order-level generative foundation models for simulating financial markets. This is an advancement research expanding on traditional statistical and agent-based models, providing promissing resolution, interactivity, and realism.

This paper is well-written, clear, and consice. It begins by identifying a specific research gap, then clearly states the proposed solution, and explains how it functions. Additionally, the paper describes potential downstream applications and further research opportunities, all within a well-organized logic chain.

**Weaknesses:**

(1) Time Interval.
The model discussed in this paper simulates market at the minute level, whereas in actual financial markets, transactions can occur at the nanosecond level. There could be challenges in accurately capturing the extreme volatility and rapid decision-making. Specifically, using a closed-source LLM engine might introduce limitations due to rate constraints. In contrast, this paper employs an open-source engine, which necessitates careful consideration of its reasoning and inferential capabilities to ensure it can effectively model the high-speed dynamics of financial markets. Does the MarS model have the design capability and flexibility to increase the level of trading time intervals?

(2) Multi-order book correlation.
In the design of large market models, the two primary approaches are the order-sequence model and the order-batch model, which capture temporal correlations and patterns. Additionally, in the conditional generation process, it seems to focus solely on a single order book. Considering that markets consist of multiple order books and each asset is significantly influenced by others as well as the market environment, how can the model effectively capture the correlations between different order books?

**Questions:**

(1) Order-sequence model
The order-sequence model uses an autoregressive approach to capture patterns within the order book, aiming to predict subsequent orders. However, the diversity of trading strategies among market participants complicates this task, often making the next order seemingly random and unpredictable. For instance, a momentum trader might buy following a price increase, anticipating continued upward movement based on the persistence of price trends. Conversely, a mean reversion trader sees a price rise as a selling opportunity, expecting prices to normalize back to the mean, counter to the current trend. Could you provide literatures or evidences in your experiments supporting a relationship between orders?

(2) Diffusion model
In Figure 1, a diffusion model is introduced. Could you explain the rationale behind this choice and how the diffusion model is utilized in this context?

---

> ### Author Response · Authors · 2024-11-21
>
> We thank the reviewer for the careful review. We address the comments below (C: comments; R: responses):
>
> **C1: Time interval and flexibility of MarS**
> - "Challenges in capturing extreme volatility and rapid decision-making."
> - "Using an open-source engine necessitates careful consideration of reasoning and inferential capabilities to ensure high-speed dynamics are accurately modeled."
>
> **R1:**
> Thank you for your insightful and professional feedback on the time interval aspect of our model. We would like to clarify that our model indeed builds and generates at the finest granularity using tick-by-tick order data. Consequently, our simulation framework is capable of modeling high-frequency trading scenarios, including nanosecond-level trading.
>
> For performance evaluation, we have chosen the minute-level metrics calculation period, which aligns with common industry practices. By observing stylized facts at the minute level and calculating minute-based simulated metrics, we effectively assess the model's performance. These minute-level observations are constructed from a series of tick-by-tick/nanosecond-level orders. The precise modeling at this microstructural level underpins the model's strong performance at the macro level.
>
> Moreover, MarS is designed to be a realistic, interactive, and controllable simulation engine, making it well-suited for handling extreme scenarios and rapid decision-making. These capabilities were integral to our design considerations, ensuring robustness and flexibility for high-frequency market dynamics.
>
> **C2: "How can MarS effectively capture correlations between multiple order books, considering market interdependencies?"**
>
> **R2:**
> Good point! Currently, the inference of MarS is set on the single asset at one time, thus only one order book will be considered. The synchronous simulation across assets and multi-order book will be our future work with new model design.

---

> > ### Comment · Reviewer_dWUs · 2024-11-26
> > **One potential consideration**
> >
> > Thank you for your response. Another aspect may worth considering is the application of the Hawkes process to model bid and ask dynamics. Essentially, this approach accounts for the influence of previous orders as well as the impact of orders on the opposite side, which are integral components of your order module.

---

> > > ### Author Response · Authors · 2024-11-26
> > >
> > > Dear Reviewer dWUs,
> > >
> > > Thank you for your insightful suggestion regarding the application of the Hawkes process. We agree that it is highly relevant to modeling bid and ask dynamics in financial markets. We will explore incorporating this approach in our future work to enhance the performance of our order module.

---

> ### Author Response · Authors · 2024-11-21
>
> **C3: "The order-sequence model uses an autoregressive approach to predict subsequent orders, but diverse trading strategies among participants make this difficult. Could you provide literature or experimental evidence supporting a relationship between orders?"**
>
> **R3:**
> Thank you for your great question. As we discussed in the Related Work section of the paper, early approaches in the field of financial market simulation indeed relied on agent-based modeling. These methods used multi-agent systems to simulate market behaviors, as described by Chiarella et al. (2009) [1], Byrd et al. (2020) [2]. However, traditional approaches fall short in learning and calibrating trader populations, as historical labeled data with details on each individual trader strategy is not publicly available.
>
> In response to these challenges, recent work by Coletta et al. (2021, 2022) [3, 4] introduced a "world agent" approach trained as a conditional GAN to learn from historical data. It is intended to emulate the overall trader population, without the need of making assumptions about individual market agent strategies. Moreover, the world agent outperforms previous multi-agent approaches based on hand-crafted market configurations, showing better market realism. Further research by the same authors [5] evaluated this method for more aspects. However, their focus is primarily on LOB modeling, they do not evaluate the model's capability to address downstream financial tasks.
>
> Our work aims to push the boundaries of financial market simulation by introducing an innovative approach. As an order-level foundation model, MarS excels in modeling the market impact of orders, achieving high levels of controllability and realism. By framing various financial market tasks as conditional trading order generation problems, we demonstrate MarS's transformative potential and practical applications in real-world financial markets.
>
> [1] Carl Chiarella, Giulia Iori, and Josep Perelló. The impact of heterogeneous trading rules on the limit order book and order flows. Journal of Economic Dynamics and Control, 33(3): 525–537, 2009.
> [2] David Byrd, Maria Hybinette, and Tucker Hybinette Balch. ABIDES: Towards high-fidelity multi-agent market simulation. In Proceedings of the 2020 ACM SIGSIM Conference on Principles of Advanced Discrete Simulation, pp. 11–22, 2020.
> [3] Andrea Coletta, Matteo Prata, Michele Conti, Emanuele Mercanti, Novella Bartolini, Aymeric Moulin, Svitlana Vyetrenko, and Tucker Balch. Towards realistic market simulations: A generative adversarial networks approach. In Proceedings of the Second ACM International Conference on AI in Finance, pp. 1–9, 2021.
> [4] Andrea Coletta, Aymeric Moulin, Svitlana Vyetrenko, and Tucker Balch. Learning to simulate realistic limit order book markets from data as a world agent. In Proceedings of the Third ACM International Conference on AI in Finance, pp. 428–436, 2022.
> [5] Andrea Coletta, Joseph Jerome, Rahul Savani, and Svitlana Vyetrenko. Conditional generators for limit order book environments: Explainability, challenges, and robustness. In Proceedings of the Fourth ACM International Conference on AI in Finance, pp. 27–35, 2023.

---

> ### Author Response · Authors · 2024-11-21
>
> **C4: "Figure 1 introduces a diffusion model. Could you explain the rationale for this choice and how the diffusion model is utilized in this context?"**
>
> **R4:**
> Thank you for your thoughtful question regarding the rationale and utilization of the diffusion model. We designed the diffusion-based generation mechanism to enhance the fine-grained controllability of MarS simulations, particularly in scenarios where vague descriptive prompts need to be translated into actionable control signals.
>
> The diffusion-based generation utilizes LLM-based historical market record retrieval to create fine-grained control signals from vague descriptive prompts. Initially, historical market records that align with the general description (DES_TEXT) are retrieved. The diffusion mechanism then generates realistic and detailed control signals (e.g., price, volume) for each time step, ensuring consistency with the retrieved historical records.
>
> In our updated design, we have further broadened the approach used for aligning with DES_TEXT. We introduce a fine-grained signal generation interface, with the diffusion-based mechanism being one of the supported methods. This interface also accepts other approaches, making it more flexible and extensible. This interface maps vague descriptions in natural language or general configurations to fine-grained series of control signals. We provide an implementation that leverages LLM-based historical market record retrieval to generate precise control signals (e.g., price, volume) from vague descriptive prompts, such as DES_TEXT. These signals guide the ensemble model, ensuring that the simulations not only follow realistic market patterns but also adhere to specific user-defined scenarios.
>
> Additionally, we have introduced this interface in Appendix E, which provides further details on how vague descriptions are mapped to fine-grained control signals.
>
> We hope this explanation clarifies the rationale and utility of the diffusion model in the context of MarS. Please let us know if further details are required.

---

> > ### Comment · Reviewer_dWUs · 2024-11-26
> > **About using diffusion model**
> >
> > Thank you for your clarification. I believe this also serves as a valuable connection between large language models that is good at summarization and diffusion model for representing numerical values.

---

### Official Review · Reviewer_nX8c · 2024-11-04

**Soundness:** 2
**Presentation:** 2
**Contribution:** 3
**Rating:** 6
**Confidence:** 4

**Summary:**

In this paper, the authors propose a financial market simulation framework, MarS, powered by a large market model (LMM) trained on real-world limit order book (LOB) data. They introduce two key algorithmic innovations: order sequence modeling, which tokenizes the LOB data, and order batch sequence modeling, which converts it into a tensor representation similar to the RGB format in image processing. These techniques connect LOB market simulation to recent advances in the field. In their experiments, the authors demonstrate that the LMM follows the scaling law for model and training data size, showcasing potential use cases for the framework.

**Strengths:**

1. A major strength of this paper is its algorithmic innovation in using order book data tokenization and an RGB-like tensor representation, which importantly bridges market simulation with advances in language and vision modeling.

2. The paper introduces a timely topic by connecting LOB market simulation with generative modeling.

**Weaknesses:**

1. The paper’s reproducibility would benefit from providing additional critical details about the method and implementation.

  - In Section 2.1.1, the authors outline that conditional trading order generation requires several hyperparameters, such as DES_TEXT for general market description and MTCH_R for specifying the matching rule for trading orders. However, the experimental section does not provide configuration details for these hyperparameters in relation to the obtained results.

  - While the authors provide sufficient detail on order sequence modeling and order batch sequence modeling, the training process for the ensemble model lacks detailed illustration, and further illustration on how the two methods are combined within the ensemble would be beneficial.

  - The authors mention that the LMM is trained on real-world market data; however, they do not provide data processing details. For example, real-world LOB data includes various order types (e.g., stop-loss orders, limit orders, market orders) that may arrive asynchronously in the market, yet the paper does not specify how these orders are mapped to the bid and ask sides as described.

2. More comprehensive evaluations are needed to demonstrate the authenticity of the simulated market. Although the authors include three tests of stylized facts, further testing to show that the framework conforms to additional stylized facts would strengthen its credibility. Please see "Cont, R. (2001). Empirical properties of asset returns: stylized facts and statistical issues. Quantitative finance, 1(2), 223." for details. Ideally,  multiple iterations and providing confidence intervals for the measures are preferred.

3. Some of the experiments are not necessarily good demonstrations about the model strength and their design need to be further improved.

  - In the 4.2 detection experiment, 1) the manipulation seems can be easily identified due to anomaly in the distribution tail without resorting to the simulation 2) the distribution similarity need to be clearly defined and demonstrate it is statistical different from the others 3) the author may want to compare with other anomaly detection algorithms.

  - The factor analysis is not a good application example because 1) it could also performed on real world data. 2) the simulated market may not good representation about the real-world market if only three stylized facts are satisfied.

4. The paper’s clarity would benefit from further improvement. Since order sequence modeling and order-batch modeling are two key innovations, the description of these methods should be included in the main content rather than in the supplementary materials.

**Questions:**

- In the section B.2, the author mentioned the index could uniquely identify a limit order. Could the authors provide a concrete example?

- In the order-batch sequence modeling part, is there any mechanism/constraint to prevent the number of cancel order be greater than the outstanding orders? If such scenario happens, how the will the framework cope with it?

- In the section 3.3, I am not following why the interaction ability is evaluated using the correlation. Could the authors elaborate?

---

> ### Author Response · Authors · 2024-11-21
>
> We thank the reviewer for the careful review. We address the comments below (C: comments; R: responses):
>
> **C1: Some configurations and hyperparameters like DES_TEXT and MTCH_R are mentioned in Section 2.1, but the experimental section does not provide configuration detail.**
>
> **R1:**
> Thank you for raising this point; we are happy to provide clarification.
>
> ---
>
> **MTCH_R (Matching Rules):**
>
> In short, our current experiments are primarily based on the double-auction rule, with detailed configurations designed to pass replay checking on the Chinese stock market. Specifically, to verify the correctness of our matching engine, we submit real historical orders to it and ensure that the generated transactions match the actual transaction data.
>
> MTCH_R represents a comprehensive set of order-matching rules, such as the widely used double auction mechanism. In real-world financial markets, these rules are specified and periodically adjusted by exchanges. In our simulation, these rules are governed by the Simulated Clearing House. We formulated MTCH_R as a hyperparameter to make the MarS framework adaptable to different markets and conditions. Expanding MTCH_R would reveal the full extent of an exchange’s trading rules, encompassing many details that we have implemented in our code for the Simulated Clearing House.
>
> Moreover, while the double auction mechanism is a common paradigm in global financial markets, there are variations in trading rules across different markets and periods. These include aspects such as price fluctuation limits, circuit breakers, and distinctions between call and continuous auction sessions. Our goal was to encapsulate these variations within the conditional trading order generation framework, ensuring the approach remains broadly applicable and flexible for different market scenarios.
>
> ---
>
> **DES_TEXT (Describe Text):**
>
> DES_TEXT is a text-based control signal and an optional configuration used in certain applications, such as what-if analysis and RL-agent training. It allows us to use textual descriptions to enable MarS to generate synthetic data under extreme or special circumstances (e.g., sharp market rises or declines). In experiments related to control signal, where DES_TEXT is an optional input, we have set this parameter to its default value, i.e., DES_TEXT = "normal market condition." In other applications like forecasting and detection, DES_TEXT is not included as an input condition.
>
> DES_TEXT is a key component in the conditional trading order generation task, acting as a control mechanism for the "conditional" aspect. Specifically, DES_TEXT is used to describe different market states under which we aim to generate trading orders. Examples of such market states include "sharp price decline" or "high market volatility." By incorporating DES_TEXT, we enable the generation process to adapt to varying market conditions, making the generated trading orders contextually relevant.
>
> In **Appendix E**, we provide a detailed example where the market state is defined as DES_TEXT = "sharply drop." This example demonstrates:
>
> 1. **Retrieval of Relevant Real-World Cases**: Using a large language model, we extract real-world data that correspond to the specified market state of "sharply drop." These retrieved cases serve as realistic instances of the described market condition.
>
> 2. **Conditional Order Generation**: Leveraging these real-world cases as control signals, the proposed MarS framework generates orders tailored to the specified market condition.
>
> This methodology highlights how DES_TEXT bridges the gap between market state descriptions and the generation of orders, ensuring that the outputs align with the intended market context.
>
> ---
>
> We hope this explanation clarifies how MTCH_R and DES_TEXT function within our framework and how they contribute to the flexibility and adaptability of MarS. We have also added an extra section in the latest manuscript to further clarify the configuration under the conditional order generation paradigm. Please check Appendix F of the new manuscript.

---

> ### Author Response · Authors · 2024-11-21
>
> **C2: Insufficient description of the ensemble model training process and how order sequence modeling and order-batch sequence modeling are combined.**
>
> **R2:**
> Thank you for your comments. Actually, we have detailed the training and generation process of the ensemble model in Appendix D, as well as in Section 2.2 and Figure 3.
>
> To clarify, the ensemble model is a simple cross-attention transformer model that is trained by taking order logits and real order channels as input to generate the next order. The model is trained using the loss advantage over the order model as a metric. As illustrated in Figure 15, the ensemble model improves its performance in order generation by conditioning on order batch data, validating the efficacy of this design.
>
> During the generation phase, the ensemble model refines the distribution of the next order using the target order-batch distribution for the subsequent minute. An order is then generated by sampling from these refined logits, becoming the latest market order. The order model and the ensemble model iteratively generate orders until the end of the next minute, ensuring a continuous and refined order generation process.
>
> We hope this explanation, along with the provided references to the relevant sections and figures, sufficiently clarifies the training and integration details of our ensemble model.
>
> **C3: Data processing details for real-world LOB data, including handling asynchronous order types (e.g., stop-loss, limit, market orders) and mapping them to bid/ask sides.**
>
> **R3:**
> Thank you for your thoughtful and professional question. We have provided implementation details regarding our data processing in Appendix B and C.
>
> In practice, various order types from brokers/end-user layers are converted into underlying-layer raw orders according to exchange trading rules. The data we receive from our provider consists of these raw orders as recorded by the exchange. For example, stop-loss orders, which are composite order types provided by brokers, are decomposed into at least two separate orders in the tick-by-tick data we receive: one recorded at the time of submission and another upon triggering the stop-loss condition, each with distinct arrival times. Limit orders and market orders are directly used in our model training. During our data processing and simulation, we strictly follow the same rules used by the exchange to handle orders, ensuring that processes such as mapping into the LOB, order matching, and execution are accurately replicated.
>
> To further alleviate any concerns, we will include additional details in the new version of the manuscript.

---

> ### Author Response · Authors · 2024-11-21
>
> **C4: Further evaluations on stylized facts in Cont, R. (2001).**
>
> **R4:**
> Thank you for the valuable suggestion. We agree that a comprehensive evaluation of stylized facts will enhance the credibility of our work. To address this, we have added a new section (Sec. I) in the appendix to evaluate Cont's 11 stylized facts[1] using both historical and simulated order sequences.
>
> To rigorously test these facts, we simulated 11,591 trajectories for the top 500 liquid stocks in the Chinese market, from March 9, 2023, to July 12, 2023. The results are summarized in the table below:
>
> | **Fact #** | **Fact Name**                                     | **Historical** | **Simulated** |
> | ---------- | ------------------------------------------------- | -------------- | ------------- |
> | 1          | Absence of autocorrelations                       | $\times$       | $\times$      |
> | 2          | Heavy tails                                       | $\times$       | $\times$      |
> | 3          | Gain/loss asymmetry                               |                |               |
> | 4          | Aggregational Gaussianity                         | $\times$       | $\times$      |
> | 5          | Intermittency                                     | $\times$       | $\times$      |
> | 6          | Volatility clustering                             | $\times$       | $\times$      |
> | 7          | Conditional heavy tails                           | $\times$       | $\times$      |
> | 8          | Slow decay of autocorrelation in absolute returns | $\times$       | $\times$      |
> | 9          | Leverage effect                                   |                |               |
> | 10         | Volume/volatility correlation                     | $\times$       | $\times$      |
> | 11         | Asymmetry in timescales                           | $\times$       | $\times$      |
>
> Key findings include:
>
> - Nine out of the 11 stylized facts are observed in both historical and simulated data. However, *Gain/loss asymmetry* and *Leverage effect* are not present, possibly reflecting modern market shifts. A recent study by Ratliff-Crain et al. (2023)[2] notes similar absences in the modern U.S. Dow 30 stocks.
>
> - All 11 facts show similar patterns between simulated and historical sequences, showcasing the model's strong capability in generating realistic order sequences.
>
> While stylized facts are key metrics to evaluate the realism of market simulation, merely evaluating stylized facts does not fully assess financial market simulation quality. Further evaluations for **in-context** generation, such as forecasting (Section 4.1) and quantitative analysis of stylized facts (Section J), are crucial.
>
> For more evaluation details, please refer to Sec. I in the revised manuscript.
>
> References:
>
> [1] Rama Cont. **Empirical properties of asset returns: stylized facts and statistical issues.** Quantitative finance, 1(2):223, 2001.
>
> [2] Ethan Ratliff-Crain, Colin M. Van Oort, James Bagrow, Matthew T. K. Koehler, and Brian F. Tivnan. **Revisiting stylized facts for modern stock markets.** In 2023 IEEE International Conference on Big Data (BigData), pp. 1814–1823, 2023. doi: 10.1109/BigData59044.2023.10386957.

---

> ### Author Response · Authors · 2024-11-21
>
> **C5: Concerns on 4.2 detection experiment ("Manipulation detection can be easily identified due to anomalies in the distribution tail without requiring simulation.", "Define distribution similarity clearly and demonstrate its statistical difference from others.", "Consider comparison with other anomaly detection algorithms.")**
>
> **R5:** Thank you for your thoughtful suggestions. We acknowledge your concerns and would like to provide additional clarification.
>
> We have defined Distribution Similarity in Appendix J: “We calculate the overlap coefficient between the empirical distribution of the stylized fact and the simulated distribution. A higher score indicates a higher similarity in the overall distribution.” In our model design for manipulation detection, we consider multiple stylized facts and evaluate declines in distribution similarity across various dimensions to detect anomalies comprehensively.
>
> In Section 4.2, the example provided serves as a representative illustration of our approach. While MarS generally performs well under normal conditions, we observe a performance drop during the manipulation period. This decline in distribution similarity can be intuitively visualized as a heavier tail and a distinct peak around a spread of 1000. It is important to note that a single anomaly does not conclusively indicate market manipulation. Instead, it serves as an initial signal that requires further holistic assessment, combining multiple metrics to ensure robust conclusions. This approach suggests a novel method for detecting market manipulation, by monitoring sudden performance drops in MarS.
>
> Our primary objective in this experiment was to demonstrate the paradigm shift MarS offers in market manipulation detection, which is why we did not include comparisons with other methods. As far as we know, traditional time-series anomaly detection techniques are insufficient for effectively addressing market manipulation. This is not only because market manipulation cannot be simply treated as a time-series anomaly detection problem, but also due to the scarcity of data samples on market manipulation, which makes it difficult to support the training of traditional machine learning methods. At present, market manipulation detection still relies heavily on extensive human expertise and requires considerable time investment.
>
> We believe that this explanation clarifies our experimental design. In the new version, we have clarified this part.
>
>
> **C6: Concerns on factor analysis ("Factor analysis could also be performed on real-world data, reducing its novelty.", "The simulated market may not adequately represent the real-world market if only three stylized facts are used.")**
>
> **R6:** Good point! We would like to address this as follows:
>
> - We agree that the factor mining approach discussed in Section 4.3 is indeed applicable to real-world data. However, we want to emphasize that conducting such analysis on real-world data often requires costly online experiments, as the data must be collected under controlled conditions with consistent time scales and trading environments. Moreover, the availability of essential features, such as Limit Order Books (LOB) and order batches, is often limited, and the majority of real-world datasets are proprietary and not publicly accessible, as highlighted in [Bacry et al., 2015], several filters have to be applied to get clean data for training. These constraints significantly limit the scope and feasibility of performing comprehensive factor analysis on real-world data.
>
> - Our primary aim is not to highlight the novelty of the factor mining method itself but rather to demonstrate the unique capabilities of MarS as a generative simulation engine. With MarS, we can easily generate a large volume of homogeneous, feature-rich trading order data that includes detailed market contexts. As mentioned in Section K of the appendix, we trained our factor mining model over 624K simulated trajectories. This allows us to explore new market impact factors and behaviors even with relatively simple factor mining techniques. The ability to simulate diverse market scenarios and extract new insights using MarS is a key contribution of this work.
>
> - We acknowledge that relying on only three stylized facts may not fully validate the reliability of MarS simulations. To address this, we have extended our validation by examining additional stylized facts, as detailed in our response to Comment 4. This broader validation strengthens the case for MarS as a robust and realistic simulation tool for financial markets.
>
> Thanks again for the valuable point, and we have also merged some responses in the new manuscript to improve the paper.

---

> ### Author Response · Authors · 2024-11-21
>
> **C7: "Would benefit from including descriptions of order sequence modeling and order-batch modeling in the main content rather than in supplementary materials."**
>
> **R7:** Good point! We have uploaded a new version of the manuscript, and highlighted the crucial details of the order model and order-batch model in Sec 2.1 of the main text.
>
> **C8: "Provide a concrete example for the index uniquely identifying a limit order, as mentioned in Section B.2."**
>
> **R8:** Thank you for your insightful question. I appreciate the opportunity to provide more details on our discretization process.
>
> To address your request for a concrete example of the index uniquely identifying a limit order, as mentioned in Section B.2, I will elaborate on how we convert real values into indices and vice versa.
>
> For order price (the price distance relative to the LOB mid-price), volume, and interval, we build a converter to discretize these values based on real data distribution. More specifically, during the converter building period, we sort the data and split it into `num_bins` slots, recording the boundary and samples of each slot. This converter then allows us to transform real values, such as volume, into the slot index. Additionally, the converter supports sampling a real value from the index, thanks to the recorded samples for each slot.
>
> Once we have the converter for price, volume, and interval, we can derive the order index from a limit order or obtain predicted order information from an index.
>
> Here is the pseudocode for the converter to get the order index from a limit order and its reverse process:
>
> ```python
> def get_order_index(
>     cur_order: LimitOrder,
>     interval_seconds: float,
>     cur_mid_price: int,
>     converter: Converter,
>     num_price_level_bins: int = 32,
>     num_order_volume_bins: int = 32,
>     num_order_interval_bins: int = 16,
> ) -> int:
>     """Get order index from order, interval and mid-price information.."""
>     type_index = ["B", "S", "C"].index(cur_order.type)  # [0, 1, 2]
>     price_slot = converter.price_level.get_bin_index(cur_order.price - cur_mid_price)
>     volume_slot = converter.order_volume.get_bin_index(cur_order.volume)
>     interval_slot = converter.order_interval.get_bin_index(interval_seconds)
>
>     order_index = (
>         type_index * num_price_level_bins * num_order_volume_bins * num_order_interval_bins
>         + price_slot * num_order_volume_bins * num_order_interval_bins
>         + volume_slot * num_order_interval_bins
>         + interval_slot
>     )
>     return order_index
>
> def get_pred_order_info(
>     order_index: int,
>     cur_mid_price: int,
>     converter: Converter,
>     num_price_level_bins: int = 32,
>     num_order_volume_bins: int = 32,
>     num_order_interval_bins: int = 16,
> ) -> PredOrderInfo:
>     """Get predicted order info, the reversed function of get_order_index."""
>     order_type = order_index // (num_price_level_bins * num_order_volume_bins * num_order_interval_bins)
>     price_slot = (order_index % (num_price_level_bins * num_order_volume_bins * num_order_interval_bins)) // (
>         num_order_volume_bins * num_order_interval_bins
>     )
>     volume_slot = (order_index % (num_order_volume_bins * num_order_interval_bins)) // num_order_interval_bins
>     interval_slot = order_index % num_order_interval_bins
>
>     pred_order_info = PredOrderInfo(
>         type=["B", "S", "C"][order_type],
>         price=cur_mid_price + converter.price_level.sample(price_slot),
>         volume=converter.order_volume.sample(volume_slot),
>         interval=converter.order_interval.sample(interval_slot),
>     )
>     return pred_order_info
> ```
> **C9: "Is there a mechanism/constraint to prevent the number of cancel orders from exceeding outstanding orders? If so, how does the framework handle this scenario?"**
>
> **R9:** Good question! MarS actually has a mechanism to prevent the number of cancel orders from exceeding the outstanding orders. Specifically, before submitting the predicted order information (as detailed in R8) to the exchange, we perform a correction step to ensure compliance with the current market state. For instance, if the model predicts a cancel volume that exceeds the available volume at a particular level, we adjust the cancel volume to match the available volume. This correction step ensures that our predictions remain realistic and executable within the constraints of the market.
>
> **C10: "Why is the interaction ability evaluated using correlation? Please elaborate."**
>
> **R10:** Thank you for your question. In Fig. 5c, we use correlation to demonstrate that including user interaction changes the order sequence, thereby decreasing its correlation with the original sequence. This indicates that our system effectively reflects user interaction. Additionally, we compare the market impact generated by user interaction with an empirical market impact model in Fig. 5b for a quantification study, showing that our system can generate realistic market impact.

---

> > ### Comment · Reviewer_nX8c · 2024-11-25
> >
> > Thank you to the authors for their thorough and detailed responses to the comments and questions. I find the work impressive, particularly in its introduction of a proxy simulation for the complex financial market. My initial comments focused on the details of the simulation system, the authenticity of the simulated market, and the reproducibility of the work, which have been largely addressed in the updated paper and the authors' responses. As a result, I plan to increase my rating to reflect these improvements.
> >
> > **R1, R3, R4, R5, R6**
> >
> > Thank you again for the detailed and thoughtful responses! I am particularly impressed by how well the simulated market aligns with the majority of stylized facts. The authors’ responses have effectively addressed my concerns and resolved the issues raised in my original comments.
> >
> > **R2**
> >
> > Thank you for the detailed explanation. From my understanding, the order sequence modeling, as shown in Figure 11, operates at a tick-level frequency, providing a fine-grained representation of market dynamics. In contrast, the order-batch model works at a more coarse-grained level, such as the minute-level aggregation depicted in Figure 14. Given that the majority of orders are generated and matched by the order-batch model, could this lead to a "bumpy" pattern in the number of orders matched, particularly due to the temporal aggregation effects of the batch approach? Additionally, is there any mechanism in place to ensure that the order sequence modeling and the order-batch model contribute roughly equally to the formation of the limit order book (LOB) shape, maintaining a balance between fine-grained and coarse-grained dynamics?

---

> ### Author Response · Authors · 2024-11-25
>
> Thank you for your positive feedback and for considering the improvements made in our work. We appreciate your plan to increase the rating and are glad to have largely addressed your concerns regarding the simulation system, the authenticity of the simulated market, and reproducibility.
>
> Regarding the concern about batch matching of orders, this can be avoided by the order **interval** generated by the ensemble model. During the generation process, the ensemble model refines the logits of orders by conditioning on the order batch. The output space remains the same as the order model (B.2.1), which includes (type, price, volume, **interval**). The **interval** here represents the time this order should arrive after its previous order, ensuring that the order sequence maintains a smooth transition, thus preventing a "bumpy" pattern in the number of orders matched.
>
> As for balancing fine- and coarse-grained information, we did not build an **explicit mechanism** to balance the order and order-batch model. On the contrary, MarS tried to merge the information and fit constraints from both the order and order-batch in an **implicit manner**, through the conditional generation process. An analogy can be drawn to video generation based on both prompts and the first frame image. In this context, the order batch acts as the prompt, containing information about future order distribution, while the existing orders act as the first frame image, representing known information. Our ensemble model strives to satisfy both the prompt (order batch) and the first frame image (order sequence), with the trade-off being implicitly managed to balance fine- and coarse-grained information.

---

> > ### Comment · Reviewer_nX8c · 2024-11-25
> >
> > Thank the authors for the response. I have no other questions for now. I have updated the scores based on the discussion. The paper makes a good contribution to the market simulation literature. Open-sourcing the code would further benefit this research community.

---

> > > ### Author Response · Authors · 2024-11-26
> > >
> > > Dear Reviewer nX8c,
> > >
> > > Thank you for your positive feedback and for updating the scores based on our discussion. We are pleased to hear that you recognize the contribution our paper makes to the market simulation literature.
> > >
> > > Regarding your suggestion to open-source the code, we agree that making the code available to the research community could significantly enhance the reproducibility and impact of our work. In fact, we have already open-sourced some core components of MarS, including the market simulation engine along with several examples. We are actively working on releasing additional code and models, ensuring they meet the necessary standards for security, usability, and documentation. We will provide an update on the code's availability in the final version of the paper.
> > >
> > > Thank you once again for your insightful comments and support.

---

### Official Review · Reviewer_CjFd · 2024-11-05

**Soundness:** 3
**Presentation:** 3
**Contribution:** 3
**Rating:** 6
**Confidence:** 4

**Summary:**

This paper makes several contributions: (i) it designs, implements and evaluates Large Market Model (LMM), an order-level generative foundation model, for financial market simulations; (ii) a novel financial Market Simulation engine (MarS), based on LMM, that enables more realistic, domain-specific interactive and controllable order generation. The paper includes evidence of LMM's strong scalability across data size and model complexity, and of MarS's advancements towards more realistic, interactive and controllable order generation with market impact. Various scenarios illustrated the application of MarS for forecasting, detection, analysis, and agent training, thus demonstrating its contributions towards more realistic financial market modelling and simulation frameworks.

**Strengths:**

The paper makes a contribution in a highly relevant application domain, i.e. realistic financial market simulation frameworks. The methodology is adequate, and experiments and results are included in support of the statements and contribution claims. The main part of the paper is focused on LLM and MaRs, presenting the design and implementation decisions, and a detailed description of the experiments and results. The paper includes evidence of the LLM's scalability across data size and model complexity. In addition, MaRs enables more realistic, interactive and controllable order generation. A range of different scenarios illustrate the applicability of MaRs to key domain-specific tasks such as forecasting, detection, analysis, and agent training, thus demonstrating its contributions towards more realistic financial market modelling and simulation frameworks.

**Weaknesses:**

The current structure of the main part of the paper prioritises a high-level presentation of LLM and MaRs, and the applicability of the proposed solution to four relevant use cases. The MaRs design and features (Section 2) are also presented at a high level, with insufficient rigour in the problem definition (Section 2.1) -- which may not be known to a ML audience. It is not clear why, for example, being able to provide a text-based description of the desired market scenario, would guarantee controllability (the definition of controllability on page 2 is fairly vague). Similarly, the assumptions made/ requirements about the user-injected orders are not specified. The language is ambiguous in places, e.g. "any possible market condition" (p. 2); "often using recent real orders" (p. 3). More technical detail could be included in Section 2.1.2, for self-contained, rigorously specified problem and solution,  and to support the claims made in this section.

It would help to better define the control signals, and discuss their effectiveness for increasing correlation scores.

**Questions:**

The main questions and suggestions are derived from the identified weaknesses.

The paper should be restructured by including more specific key definitions and design decisions, including ML-specifc (relevant for the ICLR audience) from the Supplementary Materials into the main paper.

"RL-agent training tasks" is mentioned twice in the paper (p. 8). It would help to link this term to Section 4.4, for improved clarity and readability. Also, discussing the differences between MaRs/LMM and DeepLOB would improve the paper. Clearly defining the synthetic data and synthetic market impact (rather than referring to them indirectly in the main part of paper) would also be helpful.

In addition, it would help for the authors to address reproducibility, in terms of code and data.

Overall, the suggested changes would improve the rigor, clarity and overall contribution of the paper, as well as its suitability for the ICLR audience.

There are some minor spelling/ typo errors, e.g.: "these information concludes" (p. 5); "with constrains" (p. 28); Eq.equation (p.28)

---

> ### Author Response · Authors · 2024-11-21
>
> We thank the reviewer for the careful review. We address the comments below (C: comments; R: responses):
>
> **C1: Paper structure and ML focus ("including more specific key definitions and design decisions, including ML-specific (relevant for the ICLR audience) from the Supplementary Materials into the main paper.")**
>
> **R1:** Good suggestion! We have uploaded a new version of the manuscript and moved the clarification on the order model and order-batch model from the appendix to Section 2.1.
>
> **C2: Clearly defining the synthetic data and synthetic market impact**
>
> **R2:** Good suggestion! Synthetic data refers to simulated trading order data generated using MarS for a specific asset over a defined time period. To better illustrate, replay data consists of the real historical order data of an asset over a fixed period (e.g., 15 minutes). On top of this, MarS’s interactive capabilities allow a trading agent to submit new orders based on varying strategy parameters over a shorter time frame (e.g., 5–10 minutes). These new orders interact dynamically with existing potential orders and order batches, influencing subsequent market conditions and price trajectories.
>
> Synthetic market impact is defined as the difference (or "gap") between the simulated price trajectory, influenced by the trading agent’s actions, and the real historical price trajectory. This gap quantifies the impact of the agent’s trading strategies on the market. A clear example can be seen in Figure 6(a): during the interval 9:45–9:50, the agent executes a TWAP strategy, causing observable changes in the subsequent price trajectory. The gap between the two curves represents the synthetic market impact generated by the agent’s trading actions.
>
> Based on your suggestion, we have further polished and emphasized this definition in Section 3.2 of the updated manuscript to ensure clarity and accessibility. This refined explanation is also linked to relevant visualizations (e.g., Figure 6(a)) to better illustrate the concept.
>
> **C3: The assumptions made/requirements about the user-injected orders**
>
> **R3:** We appreciate your question regarding the requirements for user-injected orders. To clarify, user-injected orders share the same format as the original orders used during our model's training. Both are structured as (type, price, volume) tuples.
>
> We explicitly distinguish "user-injected orders" from those generated by MarS, which we refer to as "background orders." This distinction is made to emphasize the user-injected orders as the primary experimental subjects interacting with the market environment. The purpose of these user-injected orders is to test the model's interaction capabilities, making it possible to analyze the system in a manner closer to the scientific paradigm of "controlled variables." By doing so, we ensure a clearer examination of the effects and interactions facilitated by MarS.
>
> We hope this clarification addresses your concern and highlights the rationale behind distinguishing these order types within our framework.
>
> **C4: Link RL-agent training task Section 4.4**
>
> **R4:** Thanks for the good suggestion! We have added the link to Section 4.4 on page 8 to improve readability in the updated manuscript.

---

> ### Author Response · Authors · 2024-11-21
>
> **C5: Differences between MarS/LMM and DeepLOB**
>
> **R5:** Thank you for your suggestion. We have created a table and created a new section (Sec. L) in the updated manuscript to highlight the differences between MarS and DeepLOB in several aspects.
>
> | Aspect           | DeepLOB                                                                                               | MarS                                                                                                                                 |
> | ---------------- | ----------------------------------------------------------------------------------------------------- | ------------------------------------------------------------------------------------------------------------------------------------ |
> | Applicable Tasks | Designed for specific forecasting tasks, trained on fixed step forecasting                            | Designed for market simulation, able to perform general forecasting through simulation, trained on next order/order batch generation |
> | Feature Input    | Uses Limit Order Book (LOB) data as input                                                             | Uses fine-grained order sequence data as input                                                                                       |
> | Model            | Relatively small, handcrafted model for LOB forecasting, hard to scale up                             | Powered by large foundation models trained on large-scale order sequence data                                                        |
> | Prediction       | Single-step prediction for fixed length forecasting (e.g., price change after 100 orders or 1 minute) | Simulation with multi-step generation                                                                                                |
>
> **C6: Reproducibility of code and data**
>
> **R6:** We appreciate your suggestion to improve the reproducibility of our work. We acknowledge the importance of this aspect and have detailed our data sources and processing procedures in Sections B.2, B.3, and C.3. Specifically, our dataset comprises the top 500 most liquid stocks in the Chinese stock market, spanning from 2017 to 2023 and containing 16 billion order tokens. This dataset consists of standardized commercial data acquired from data vendors, and while it is publicly available for purchase.
>
> Due to the proprietary nature of this commercial dataset, we are unable to make the full data publicly accessible. However, we plan to provide downloadable data samples where possible, which will help facilitate further research. Additionally, we are committed to releasing the core code to support reproducibility.
>
> While we are eager to release our data and model specifications, security implications must be considered. The MarS, as a foundational model with potential industry-wide impact, requires careful handling to prevent abuse. We are evaluating strategies to mitigate these risks while balancing the need for openness and collaboration.
>
> **C7: Minor language and formatting issues**
>
> **R7:** We appreciate the detailed check and have already fixed them in the latest version of the manuscript.

---

> > ### Author Response · Authors · 2024-11-27
> > **further question and feedback**
> >
> > Dear reviewer CjFd,
> >
> > The revision deadline is approaching, and we sincerely hope to receive any feedback or questions you may have beforehand. If there are any issues, please let us know at your earliest convenience so we can address them promptly.
> >
> > Best regards,
> >
> > The Authors

---

> > > ### Comment · Reviewer_CjFd · 2024-11-27
> > >
> > > Thank you for the detailed and extensive responses to my and other reviewers' comments, and subsequent changes to the paper. I have increased my score to reflect this.
> > >
> > > I think the paper now represents a novel and sound contribution, relevant to the ICLR community.
> > >
> > > In terms of potential directions for improvement, controllability is still not defined rigorously, but nevertheless claimed throughout the paper (e.g. "DES_TEXT: A general description of the desired market scenario (e.g., “price bump” or “volatility crush”), ensuring controllability", p. 4) . It would also help to have a more meaningful discussion of the two stylized facts - Gain/loss asymmetry, and Leverage effect that were not matched. As also mentioned by the other reviewers, reproducibility has only been partially addressed. The paper still seems to have prioritised breadth - in terms of functionality and features, over depth and specification detail and readability (for example in the level of detail provided for diffusion, definition of controllability, RL features evaluation). Doing the latter would have made the paper more accessible to the ML community. Some of the statements, such as: "MarS provides the tools needed to explore any possible market condition" (p. 2)  or "This feature is vital for analyzing trading strategies, managing systemic risks, and developing regulatory policies in a controlled, risk-free environment" (p. 3),  are insufficiently supported by evidence/large scale-experiments, results and critical evaluation. I think it would help to include the authors' statement: "At this stage, MarS is designed to model single assets and their corresponding Limit Order Books (LOB) during inference. This single-asset focus constrains the system's ability to capture cross-asset interactions, such as arbitrage opportunities or correlated trading behaviors. Addressing this limitation by extending MarS to simultaneously model multiple assets and their interactions is an important goal for future iterations." in the paper, for completeness and clarity.
> > >
> > > There are still some minor typos/ grammar issues, which could be easily addressed: "with constrain" (p.34)/ TEXT_DES instead of DES_TEXT.

---

> ### Author Response · Authors · 2024-11-28
>
> Dear Reviewer CjFd,
>
> Thank you for your thoughtful follow-up comments and for recognizing the improvements made to the paper. We greatly appreciate your feedback and the valuable directions it provides for further enhancement.
>
> - **Controllability**: We agree that a more rigorous definition would strengthen the paper. This is a priority for future work, where we plan to quantify and evaluate controllability across diverse scenarios using formal metrics.
> - **Discussion on the Absence of Two Stylized Facts**: Thank you for the insightful suggestion. We agree that a more meaningful discussion on the absence of the stylized facts—Gain/Loss Asymmetry and the Leverage Effect—would enhance the paper. The existing study by Ratliff-Crain et al. (2023)[1] suggests that similar absences in their study are primarily due to modern market shifts. As they note, "the technological arms race and resulting market fragmentation in the intervening decades since Cont’s study fundamentally changed the dynamics of the U.S. stock market." To gain more confident evidence, it would be necessary to test these stylized facts across different periods. This would help in understanding the impact of technological advances and regulatory changes over time. We are enthusiastic about exploring this direction further in future research.
> - **Reproducibility and Depth**: We have open-sourced some core components of MarS, including the market simulation engine and examples. Additional code and models are being prepared for release, with a focus on security, usability, and documentation. Updates on availability will be included in the final version.
> - **Single-Asset Limitation**: Your suggestion to include the statement on MarS’s single-asset limitation and implications for cross-asset interactions is excellent. We will incorporate it in future iterations, supported by further evaluation. Typos have been noted and will be corrected—thank you for detailed checking and pointing them out!
>
> Lastly, we deeply appreciate your mention of an increased score, though we currently do not see this reflected in the system. **Could you kindly double-check if the score adjustment was updated?**
>
> Thank you again for your constructive feedback and recognition of MarS’s contributions. We look forward to continuing to refine and improve this work.
>
> Reference:
>
> [1] Ethan Ratliff-Crain, Colin M. Van Oort, James Bagrow, Matthew T. K. Koehler, and Brian F. Tivnan. Revisiting stylized facts for modern stock markets. In 2023 IEEE International Conference on Big Data (BigData), pp. 1814–1823, 2023. doi: 10.1109/BigData59044.2023.10386957.

---

### Author Response · Authors · 2024-11-21
**Summary of Rebuttal**

We sincerely thank the reviewers for their thoughtful and constructive feedback, which has helped us significantly improve the quality and clarity of our paper. Across the reviews, several strengths were consistently highlighted:

- **Realism and Interactivity**: Both Reviewer CjFd and Reviewer dWUs commended MarS for advancing financial market simulation with high realism, interactivity, and broad applicability to tasks like forecasting and anomaly detection.
- **Algorithmic Innovation**: Reviewers nX8c and dx5Y praised the innovative use of order book tokenization, RGB-like tensor representation, and the LLaMA2 architecture, showcasing MarS’s scalability and its alignment with advances in generative modeling.

These acknowledgments reinforce MarS’s reliability and impact. Guided by this feedback, we have made significant improvements, summarized below.

Below, we summarize the key points raised in the reviews and outline the corresponding improvements made to the manuscript.

### 1. **Structural and Presentation Improvements**

We have updated the manuscript to enhance its structure and presentation, particularly focusing on problem formulation and representation learning.

- **Tokenization of Orders and Order-Batches**: We emphasized this concept in the main paper to better align with the interests of the ML audience. To achieve this, we moved several critical discussions from the appendix into the main text, making the content more accessible and relevant.
- **Marked Additions**: For transparency, all new content is highlighted in blue in the revised manuscript. This includes expanded explanations in Sections 2.1 and 2.1.1, where we discuss tokenization, the order model, and the ensemble model in greater depth, Technical Clarity, and extra Evaluations, discussed below.

### 2. **Technical Clarity**

We have incorporated more detailed discussions of our methods and technical aspects, addressing key reviewer concerns:

- **Expanded Sections**:
  - **Section 2.1**: We added detailed explanations on tokenization, the order model, the order-batch model, and the ensemble model.
  - **Section 2.1.1 and new section F**: This now includes a thorough discussion of DES_TEXT and MATCH_R. Additionally, we have added a new 2-page appendix section (Section F, pp. 26–27) to provide a detailed overview of configuration settings across different applications.
- **Market Impact Clarifications**: Sections 3.2 and 4.3 have been expanded to provide more nuanced discussions and clarifications on synthetic market impact and its long-term dynamics.

### 3. **Additional Evaluations**

We carefully considered the reviewers' suggestions for further evaluations, particularly regarding stylized facts.

- **11 Stylized Facts**: We evaluated MarS against 11 stylized facts following Cont (2001) and Ratliff-Crain et al. (2023), finding that all exhibited similar patterns between simulated and historical sequences. The results are summarized in the table below:

| **Fact #** | **Fact Name** | **Historical** | **Simulated** |
| -| -| -| -|
| 1 | Absence of autocorrelations | $\times$  | $\times$      |
| 2 | Heavy tails | $\times$ | $\times$ |
| 3 | Gain/loss asymmetry |  | |
| 4 | Aggregational Gaussianity | $\times$ | $\times$ |
| 5 | Intermittency  | $\times$ | $\times$ |
| 6 | Volatility clustering | $\times$ | $\times$ |
| 7 | Conditional heavy tails  | $\times$ | $\times$      |
| 8 | Slow decay of autocorrelation in absolute returns | $\times$  | $\times$ |
| 9 | Leverage effect | | |
| 10 | Volume/volatility correlation | $\times$ | $\times$ |
| 11 | Asymmetry in timescales  | $\times$ | $\times$ |

Table: Presence of Stylized Facts in Historical and Simulated Order Sequences. All facts are present in both historical and simulated data, except for *Gain/loss asymmetry* and *Leverage effect*.

- **New Section**: We added a detailed 3.5 pages Section I to the updated manuscript, presenting the results of these evaluations. We welcome reviewers’ further feedback on this new analysis.

### 4. **Reproducibility and Open Source**

We recognize the importance of reproducibility and transparency. While the proprietary nature of our data limits full public access, we have taken the following steps:

- **Data Details**: The manuscript now provides extensive details on data sources, preprocessing, and configurations (Sections B.2, B.3, and C.3).
- **Open-Source Plans**: We will release core code and provide downloadable data samples to facilitate further research. However, as MarS has significant industry-wide implications, we are carefully evaluating strategies to balance openness with the need to prevent abuse.

We hope these updates address the reviewers' concerns comprehensively and demonstrate our commitment to improving the quality and impact of our work. We sincerely thank the reviewers again for their time and valuable suggestions, which have greatly enriched this research.

---

### Author Response · Authors · 2024-12-02
**Acknowledgment and Newly Created Anonymous Open-Source Repository**

We sincerely appreciate the reviewers' recognition of our paper's improvements and their invaluable feedback, which has significantly enhanced our work and inspired future research directions.

In response to the reviewers' comments on open-sourcing, we have created an anonymous GitHub repository: MarS-anonymous[1]. This repository, anonymized for the review process, includes key components of MarS, such as the core market simulation engine and examples. The correctness of the market simulation engine has been validated through extensive tests on billions of order matches, providing an efficient and robust tool for market simulation. We target to release additional code and models in the next few months, ensuring they meet the necessary standards for security, usability, and documentation. Updates on the code's availability will be provided in the final paper.

Thank you once again for your valuable feedback and support.

Reference:

[1]: MarS's anonymous GitHub repo, https://anonymous.4open.science/r/MarS-anonymous-A321/README.md

---

### Meta-Review · Area_Chair_brYA · 2024-12-20

**Metareview:**

This paper describes a generative model for simulating financial market data. All reviewers found the paper to be addressing an interesting and important topic and that it was well argued and supported.

**Additional Comments On Reviewer Discussion:**

Looks like one of the reviewers increased their score during the discussion.

---

### Decision · Program_Chairs · 2025-01-22

Accept (Poster)